# Exploration of PM$_{2.5}$ sources on the regional scale in the Pearl River Delta based on ME-2 modeling

Xiao-Feng Huang[1], Bei-Bing Zou[1], Ling-Yan He[1], Min Hu[2], André S. H. Prévôt[3], Yuan-Hang Zhang[2]

[1]Key Laboratory for Urban Habitat Environmental Science and Technology, School of Environment and Energy, Peking University Shenzhen Graduate School, Shenzhen, 518055, China.

[2]State Key Joint Laboratory of Environmental Simulation and Pollution Control, College of Environmental Sciences and Engineering, Peking University, Beijing, 100871, China.

[3]Paul Scherrer Institute (PSI), 5232 Villigen-PSI, Switzerland.

**Abstract:**

The Pearl River Delta (PRD) of China, which has a population of more than 58 million people, is one of the largest agglomerations of cities in the world and had severe PM$_{2.5}$ pollution at the beginning of this century. Due to the implementation of strong pollution control in recent decades, PM$_{2.5}$ in the PRD has continuously decreased to relatively lower levels in China. To comprehensively understand the current PM$_{2.5}$ sources in the PRD to support future air pollution control strategy in similar regions, we performed regional-scale PM$_{2.5}$ field observations coupled with a state-of-the-art source apportionment model at six sites in four seasons in 2015. The regional annual average PM$_{2.5}$ concentration based on the 4-month sampling was determined to be 37 μg/m³, which is still more than three times the WHO standard, with organic matter (36.9%) and SO$_4^{2-}$ (23.6%) as the most abundant species. A novel multilinear engine (ME-2) model was firstly applied to a comprehensive PM$_{2.5}$ chemical dataset to perform source apportionment with predetermined constraints, producing more environmentally meaningful results compared to those obtained using traditional positive matrix factorization (PMF) modeling. The regional annual average PM$_{2.5}$ source structure in PRD was retrieved to be secondary sulfate (21%), vehicle emissions (14%), industrial emissions (13%), secondary nitrate (11%), biomass burning (11%), secondary organic aerosol (SOA, 7%), coal burning (6%), fugitive dust (5%), ship emissions (3%) and aged sea salt (2%). Analyzing the spatial distribution of PM$_{2.5}$ sources under different weather conditions clearly identified the central PRD area as the key emission area for SO$_2$, NOx, coal burning, biomass burning, industrial emissions and vehicle emissions. It was further estimated that under the polluted northerly air flow in winter, local emissions in the central PRD area accounted for approximately 45% of the total PM$_{2.5}$, with secondary nitrate and biomass burning being most abundant; in contrast, the regional transport from outside the PRD accounted for more than half of PM$_{2.5}$, with secondary sulfate representing the most abundant transported species.

**Keywords:** source apportionment; ME-2; local emissions; regional transport; Pearl River Delta.

*Correspondence to*: L.-Y. He (hely@pku.edu.cn).

## 1 Introduction

With China's rapid economic growth and urbanization, air pollution has become a serious problem in recent decades. Due to its smaller size, fine particulate matter ($PM_{2.5}$) can carry toxic chemicals into human lungs and bronchi, causing respiratory diseases and cardiovascular diseases that can harm human health (Sarnat et al., 2008; Burnett et al., 2014). In particular, long-term exposure to high concentrations of fine particulate matter can also lead to premature death (Lelieveld et al., 2015). The Chinese government has attached great importance to improving air quality and issued the "Air Pollution Prevention and Control Action Plan" in September 2013, clearly requiring the concentrations levels of fine particulate matter in a few key regions, including the Pearl River Delta (PRD), to drop by 2017 from 15 to 25% of their values in 2012. The Pearl River Delta is one of the fastest-growing regions in China and the largest urban agglomeration in the world; it includes the cities of Guangzhou, Shenzhen, Zhuhai, Dongguan, Foshan, Huizhou, Zhongshan, Zhaoqing and Jiangmen, and contains more than 58 million people. The $PM_{2.5}$ concentration in this region reached a high level of 58 μg/m³ in 2007 (Nanfang Daily, 2016); however, the air quality has significantly improved due to the implementation of strict air pollution control measures, which occurred here earlier than in other regions in China. The annual average concentration of $PM_{2.5}$ in the PRD dropped to 34 μg/m³ in 2015 (Ministry of Environmental Protection, 2016).

In recent years, the receptor model method (commonly, positive matrix factorization) in the PRD was applied to perform the source apportionment of $PM_{2.5}$, which was carried out in several major cities, including Guangzhou (Gao et al., 2013; Liu et al., 2014; Wang et al., 2016), Shenzhen (Huang et al., 2014b), Dongguan (Wang et al., 2015; Zou et al., 2017) and Foshan (Tan et al., 2016). However, the above source apportionment studies only focused on part of $PM_{2.5}$ (e.g., organic matter) or single city in PRD (e.g., Shenzhen and Dongguan), lacking the extensive representation of the PRD region in terms of simultaneous sampling in multiple cities. Since the lifetime of $PM_{2.5}$ in the surface layer of the atmosphere is days to weeks and the cities in PRD are closely linked, the transport of $PM_{2.5}$ between cities should be specifically noteworthy (Hagler et al., 2006). On the other hand, although the positive matrix factorization (PMF) model has been successfully applied to source apportionment in the PRD, the apportionment with PMF has high rotational ambiguity and can output non-meaningful or mixed factors. Under such conditions, the multilinear engine (ME-2) model can guide the rotation toward a more objective optimal solution by utilizing a priori information (i.e., predetermined factor profiles). In recent years, ME-2, initiated and controlled via the Source Finder (SoFi) written by the Paul Scherrer Institute, was successfully developed to apportion the sources of organic aerosols (Canonaco et al., 2013). The novel ME-2 model has become a widely used and successful source analysis technique (e.g. Crippa et al., 2014; Fröhlich et al., 2015; Visser et al., 2015; Elser et al., 2016; Reyes-Villegas et al., 2016). The key challenges in running ME-2 are the construction of the appropriate constraint source profiles and the determination of factor numbers, and PMF could serve as the first step when using ME-2 for the determination of the priori information needed.

Accurately understanding the regional characteristics of $PM_{2.5}$ sources in the PRD can certainly guide the regional joint prevention and control of $PM_{2.5}$ in this region and provide useful references for future air pollution control strategies in China. Thus, in this study, the $PM_{2.5}$ mass and chemical compositions were measured during four seasons in 2015 at six sites in the PRD, which basically represent the pollution level of the PRD on a regional scale rather than on a city

scale. For the first time, the novel ME-2 model via the SoFi was applied to a comprehensive chemical dataset (including EC, OM, inorganic ions and metal elements) to identify the sources of bulk $PM_{2.5}$ in the regional scale of PRD; then, the spatial locations of the sources were systematically explored using the analysis of weather conditions.

## 2 Experimental methodology

### 2.1 Sampling and chemical analysis

The PRD is located in south central Guangdong Province. Based on the layout of the cities in the PRD, six sampling sites were selected to represent urban, suburban, and background sites. Detailed descriptions of these sampling sites are listed in Table 1, and their locations are shown on the regional map in Fig. 1.

**Table 1.** Description of the sampling sites in the PRD.

| Site | Site code | Coordinates | Site description | |
|------|-----------|-------------|------------------|---|
| Doumen | DM | Lat: N 22.23 | Suburban | Contains industrial areas |
| | | Lon: E 113.30 | | |
| Qi-Ao island | QA | Lat: N 22.43 | Background | An area for eco-tourism |
| | | Lon: E 113.63 | | |
| Heshan | HS | Lat: N 22.73 | Suburban | Contains industrial areas and farmlands |
| | | Lon: E 112.93 | | |
| Modiesha | MDS | Lat: N 23.11 | Urban | Contains dense urban traffic |
| | | Lon: E 113.33 | | |
| University Town | UT | Lat: N 22.59 | Urban | Contains urban traffic |
| | | Lon: E 113.98 | | |
| Dapeng | DP | Lat: N 22.63 | Background | An area for eco-tourism |
| | | Lon: E 114.41 | | |

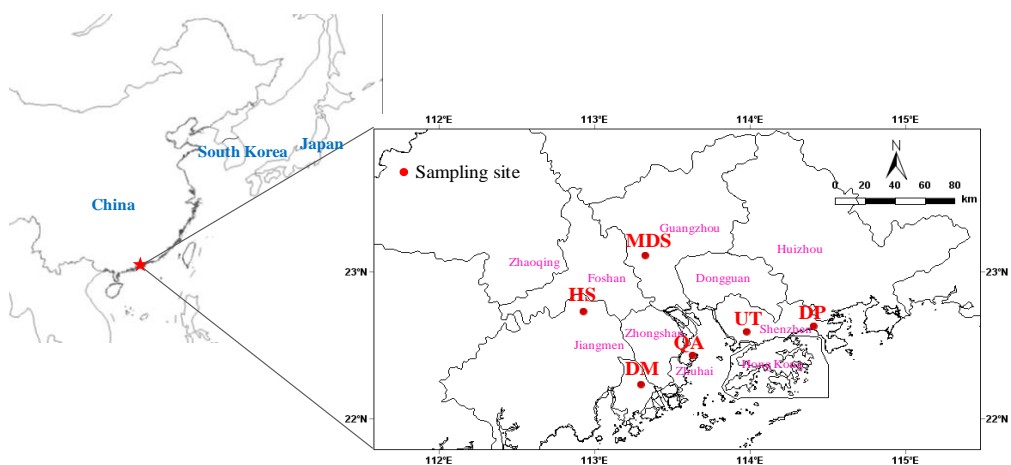

**Fig. 1.** Spatial distribution of the sampling sites in the PRD.

Samples were collected every other day during a one-month long period for each season in 2015, and Table 2 contains the detailed sampling information to refer to. Each sampling period lasted for 24 h at each site. The sampling sites of University Town (UT) and Dapeng (DP) used Thermo 2300 $PM_{2.5}$ samplers (Thermo Fisher Scientific Inc., Waltham, Massachusetts, USA, with

a flowrate of 16.7 L/min for two channels and a flowrate of 10.0 L/min for the other two channels), while those in Modiesha (MDS), Heshan (HS), Qi-Ao Island (QA) and Doumen (DM) used TH-16A $PM_{2.5}$ samplers (Tianhong Corp., Wu Han, China, with a flow rate of 16.7 L/min for four channels). Prior to the sampling campaigns, the six samplers used sampled in parallel for three times, and each time lasted for 12 h. The standard deviation of the $PM_{2.5}$ mass concentrations obtained by the six samplers in each parallel sampling was within 5%. After each sampling, the Teflon filters were put into Poly tetra fluoroethylene (PTFE) boxes and the Quartz filters were put into PTFE boxes with 500 ℃ burned aluminum foil inside. The sample boxes were then sealed by Parafilm, stored in an ice-packed cooler during transportation, and stored under freezing temperatures before analysis. A total of 362 valid samples (15-16 samples at each site for each season) were collected in this study. In addition, to track the possible contamination caused by the sampling treatment, a field blank sample was collected at each site for each season. The $PM_{2.5}$ mass can be obtained based on the difference in the weight of the Teflon filter before and after sampling in a cleanroom at conditions of 20℃ and 50% relative humidity, according to the QA/QC procedures of the National Environmental Protection Standard (NEPS, MEE, 2013b). The Teflon filters were analyzed for their major ion contents ($SO_4^{2-}$, $NO_3^-$, $NH_4^+$ and $Cl^-$) via an ion chromatography system (ICS-2500, Dionex; Sunnyvale, California, USA), following the guidelines of NEPS (MEE, 2016a, b). The metal element contents (23 species) were analyzed via an inductively coupled plasma mass spectrometer (ICP-MS, auroraM90; Bruker, Germany), also following the guidelines of NEPS (MEE, 2013a). The Quartz filters were analyzed for organic carbon (OC) and elemental carbon (EC) contents using an OC/EC analyzer (2001A, Desert Research Institute, Reno, Nevada, USA), following the IMPROVE protocol (Chow et al., 1993). The overall organic mass (OM) was estimated as 1.8 ✕ OC. In previous aerosol mass spectrometer (AMS) measurement for $PM_1$, the OM/OC ratio was measured to be 1.6 for urban atmosphere (He et al., 2011) and 1.8 for rural atmosphere (Huang et al., 2011). We adopted a uniform OM/OC ratio of 1.8 in this study because it is assumed that the mass difference between $PM_1$ and $PM_{2.5}$ may mostly contain aged regional aerosol with higher OM/OC.

**2.2 Meteorological conditions and weather classification**

The meteorological conditions during the observation period, shown in Table 2, indicated that the PRD region experienced a hot and humid summer and a cool and dry winter, while spring and fall were two transition seasons. Furthermore, the back trajectories of the air masses obtained using the NOAA HYSPLIT model (Fig. S1) revealed that the air masses originated from the northern inland in winter, from the northern inland and the South China Sea in spring, from the South China Sea in summer, and from the northeast coast and the northern inland in fall.

**Table 2.** General meteorological conditions during the observation period in the PRD.

| | Mean Temp. (℃) | Rainfall (mm) | Mean RH (%) | Mean wind speed (m/s) | Predominant wind direction |
| --- | --- | --- | --- | --- | --- |

| | | | | | |
|---|---|---|---|---|---|
| Winter (Jan.10-Feb.9) | 17 | 35 | 63% | 2.1 | ENE |
| Spring (Apr.2-Apr.30) | 23 | 61 | 72% | 1.8 | SSW |
| Summer (Jul.1-Jul.29) | 29 | 244 | 74% | 2.1 | SW |
| Fall (Oct.11-Nov.10) | 25 | 92 | 68% | 1.7 | NNE |

Changes in meteorological conditions with the seasons have significant influences on the air
quality in the PRD (Hagler et al., 2006). The same type of weather is often repeated. Physick et al.
(2001) classified the weather over the region surrounding Hong Kong into seven categories based
on surface pressure patterns, i.e., as northerly (winter monsoon), northeasterly (winter monsoon),
easterly or southeasterly, trough, southerly or southwesterly (summer monsoon), cyclonic 1 and
cyclonic 2 weather types. The PRD region, including Hong Kong, has nearly the similar weather
patterns and similar meteorological conditions. In this study, the daily weather types during the
observation period (excluding rainy days) were also classified into seven categories based on
surface pressure patterns. However, according to the surface horizontal wind vectors, the PRD was
mostly impacted by two types of airflow, i.e., southerly flow and northerly flow. Southerly flow,
including the southeasterly and southerly or southwesterly (summer monsoon) weather types, was
relatively clean and originated from the ocean (e.g., Fig. S2 and Fig. S4). Northerly flow,
including the northerly (winter monsoon) and northeasterly (winter monsoon) weather types, was
relatively polluted and originated from the north mainland (e.g., Fig. S3 and Fig. S5). Southerly
flow and northerly flow appeared with the highest frequency in the PRD (i.e., above 80%),
followed by cyclone (10%), easterly (2%) and trough (2%). In this study, southerly flow days
(PM$_{2.5}$ ≤ 17 μg/m³, see Table 3) were selected to better reflect the local source regions in the PRD,
and northerly flow days (PM$_{2.5}$ ≥ 75 μg/m³, see Table 3) were selected to better understand the
pollution accumulation process and regional transport characteristics of pollutants in the PRD. The
sampling days for southerly flow and northerly flow are listed in Table 3.

**Table 3.** Sampling days categorized as southerly flow and northerly flow days.

| Southerly flow | Wind speed (m/s) | PM$_{2.5}$ (μg/m³) | Northerly flow | Wind speed (m/s) | PM$_{2.5}$ (μg/m³) |
|---|---|---|---|---|---|
| 2015.07.01 | 2.6 | 16 | 2015.01.18 | 2.3 | 78 |
| 2015.07.03 | 3.6 | 17 | 2015.01.20 | 1.5 | 82 |
| 2015.07.15 | 1.9 | 17 | 2015.02.03 | 2 | 75 |
| 2015.07.23 | 2.6 | 12 | 2015.02.07 | 1.7 | 101 |
| 2015.07.25 | 2 | 13 | 2015.02.09 | 2.2 | 75 |
| 2015.07.29 | 1.3 | 12 | | | |


**2.3 Input data matrices for source apportionment modeling**
PMF is a multivariate factor analysis tool widely used for aerosol source apportionment. The
PMF algorithm groups the measured matrix **X** (Eq. (1)) into two non-negative constant matrices **G**
(factor time series) and **F** (factor profiles), and **E** denotes the model residuals (Paatero and Tapper,
1994). The entries in **G** and **F** are fitted using a least-squares algorithm that iteratively minimizes
the object function $Q$ in Eq. (2), where $e_{ij}$ are the elements of the residual matrix **E**, and $u_{ij}$ are
the errors/uncertainties of the measured species $x_{ij}$.
$$\mathbf{X} = \mathbf{G} \cdot \mathbf{F} + \mathbf{E} \tag{1}$$
$$Q = \sum_{i=1}^{n} \sum_{j=1}^{m} \left( e_{ij}/u_{ij} \right)^2 \tag{2}$$
The multilinear engine (ME-2) was later developed by Paatero (1999) based on the PMF
algorithm. In contrast to an unconstrained PMF analysis, ME-2 can utilize the constraints (i.e.,
predetermined factor profiles) provided by the user to enhance the control of rotation for a more
objective solution. One or more factor profiles can be expediently input into ME-2, and the output
profiles are allowed to vary from the input profiles to some extent. When using ME-2 modeling,
the "mixed factors" can usually be better resolved.
In this study, both PMF and ME-2 models were run for the datasets observed in the PRD. We
first need to determine the species input into the models. Species that may lead to high species
residuals or lower $R^2$ values between measured and model-predicted or non-meaning factors were
not included, such as those that fulfilled the following criteria: (1) species that were below
detection in more than 40% of samples; (2) species that yielded $R^2$ values of less than 0.4 in
inter-species correlation analysis; and (3) species that had little implication for pollution sources
and lower concentrations. Therefore, 18 species were input into the models; these species
accounted for 99.6% of the total measured species and included OM, EC, $SO_4^{2-}$, $NO_3^-$, $NH_4^+$, $Cl^-$,
K, Ca, Na, Mg, Al, Zn, Fe, Cd, V, Ni, Ti and Pb.
The application of PMF or ME-2 also depends on the estimated realistic uncertainty ($u_{ij}$) of
the individual data point of an input matrix, which determines the $Q$ value in Eq. (2). Therefore,
the estimation of uncertainty is an important component of the application of these models. There
are many sources of uncertainty, including sampling, handling, transport, storage, preparation, and
testing (Leiva et al., 2012). In this study, the sources of uncertainty that contributed little to the
total uncertainty could be neglected, such as replacing filters, sample transport and sample storage
under the strict QA/QC. Therefore, we first considered the uncertainties introduced by sampling
and analysis processes, such as sampling volume, repeatability analysis and ion extraction. The
species uncertainties $u_{ij}$ are estimated using Eq. (5), where $\bar{u}_c$ is the error fraction of the species,
which is estimated using the relative combined error formula Eq. (6) (BIPM et al., 2008).
$$u_{ij} = \bar{u}_c \times x_{ij} \qquad (5)$$
$$\bar{u}_c = \sqrt{\bar{u}_f^2 + \bar{u}_r^2 + \bar{u}_e^2} \qquad (6)$$
*where $\bar{u}_f$ is the relative error of the sampling volume; $\bar{u}_r$ is the relative error of the repeatability*
*analysis of the standard species; and $\bar{u}_e$ is the relative error of the ion extraction of multiple*
samples. When the concentration of the species is below the detection limit (DL), the
concentration values were replaced by 1/2 of DL, and the corresponding uncertainties were set at
5/6 of DL. Missing values were replaced by the geometric mean of the species with corresponding
uncertainties of 4 times their geometric mean (Polissar et al., 1998). The uncertainties of $SO_4^{2-}$,
$NH_4^+$ and all metal elements, which have scaled residuals larger than $\pm 3$ due to the small
analytical uncertainties, need to be increased to reduce their weights in the solution (Norris et al.,
2014). In addition, the uncertainties of EC caused by pyrolyzed carbon (PC), the uncertainties of
OM, $NO_3^-$ and $Cl^-$ due to semi-volatility under high ambient temperatures should also be taken
into account (Cao et al., 2018). In this study, more reasonable source profiles can be obtained
when further increasing the estimated uncertainties ($\bar{u}_c$) of all species by a factor of 2.
**2.4 Constraint setup in ME-2 modeling**
In this study, the USEPA PMF v5.0 was applied with the concentration matrix and
uncertainties matrix described above to identify the $PM_{2.5}$ sources. After examining a range of
factor numbers from 3 to12, the nine-factor solution output by the PMF base run ($Q_{true}/Q_{exp}$=2.5)
was found to be the optimal solution, with the scaled residuals approximately symmetrically
distributed between –3 and +3 (Fig. S6) and the most interpretable factor profiles (Fig. S7). The
model-input total mass of the 18 species and the model-reconstructed total mass of all the factors
showed a high correlation ($R^2$=0.97, slope=1.01) (Fig. S8). The factor of biomass burning was not
extracted in the eight-factor solution, while the factor of fugitive dust was separated into two
non-meaningful factors when more factors were set to run PMF. For the nine-factor solution of
secondary sulfate-rich, secondary nitrate-rich, aged sea salt, fugitive dust, biomass burning,
vehicle emissions, coal burning, industrial emissions and ship emissions, the source judgment
based on tracers for each factor was identical to that of the ME-2 results detailed in Section 3.2.
However, in Fig. S7, some factors seemed to be mixed by some unexpected components and were
thus overestimated. For example, the secondary sulfate-rich and secondary nitrate-rich factors of
PMF had certain species from primary particulates, such as EC, Zn, Al, K and Fe, among which
EC had obvious percentage explained variations (EV) values of 18.7% and 9.7%, respectively;
the EV value of OM in the sea salt factor (which was theoretically negligible) had a high value of
6.4%, and OM accounted for 37% of the total mass of this factor; the EV value of $SO_4^{2-}$ in the
fugitive dust factor (which was theoretically negligible) had a high value of 8.6%, and the $SO_4^{2-}$
concentration accounted for 26% of the total mass of this factor.
SoFi is a user-friendly interface developed by PSI for initiating and controlling ME-2
(Canonaco et al., 2013), and it can conveniently constrain multiple factor profiles. Although
USEPA PMF v5.0 can also use some priori information (such as ratio of elements in factor) to
control the rotation after the base run, it is not able to use multiple constrained factor profiles to
control the rotation (Norris et al., 2014). Therefore, SoFi is a more convenient and powerful tool
to establish various constrained factors for source apportionment modeling. Using the same
species concentration matrix and uncertainties matrix, we ran the ME-2 model via SoFi for 9–12
factors with the four factors constrained as described above, as shown in Table 4. The following
considerations were used. Secondary sulfate and secondary nitrate factors should theoretically not
contain species from primary particulates, but they may contain secondary organic matter related
to the secondary conversion process of $SO_2$ and NOx (He et al., 2011; Yuan et al., 2006b; Huang
et al., 2014b). Therefore, the contributions of the species from primary particulates were
constrained to zero in the input secondary aerosol factors, while others were not constrained. In
addition, the factors of sea salt and fugitive dust in primary aerosols could be understood based on
the abundance of species in seawater and the upper crust (Mason, 1982; Taylor and Mclennan,
1995). As seen in Table S1, the abundances of $Cl^-$, $Na^+$, $SO_4^{2-}$, $Mg^{2+}$, $Ca^{2+}$ and $K^+$ in sea salt were
relatively high, as were the abundances of Al, Fe, Ca, Na, K, Mg and Ti in fugitive dust. Therefore,
these high-abundance species were not constrained in the sea salt and fugitive dust factors, while
the other species (with abundances of less than 0.1% in the particulates) were constrained to zero
(Table 4). In addition, $HNO_3$ might react with sea salt to displace $Cl^-$ (Huang et al., 2006); thus,
$NO_3^-$ was also not constrained in the sea salt factor.
**Table 4.** The constraints of factor species for ME-2 modeling.

| Factors | OM | EC | $Cl^-$ | $NO_3^-$ | $SO_4^{2-}$ | $NH_4^+$ | Ca | Ti | V | Ni | Zn | Cd | Pb | Na | Mg | Al | K | Fe |
|---|---|---|---|---|---|---|---|---|---|---|---|---|---|---|---|---|---|---|
| Secondary sulfate | − | 0 | 0 | 0 | − | − | 0 | 0 | 0 | 0 | 0 | 0 | 0 | 0 | 0 | 0 | 0 | 0 |

| | | | | | | | | | | | | | | | | | | |
|---|---|---|---|---|---|---|---|---|---|---|---|---|---|---|---|---|---|---|
| Secondary nitrate | − | 0 | 0 | − | 0 | − | 0 | 0 | 0 | 0 | 0 | 0 | 0 | 0 | 0 | 0 | 0 | 0 |
| Sea salt | 0 | 0 | − | − | − | 0 | − | 0 | 0 | 0 | 0 | 0 | 0 | − | − | 0 | − | 0 |
| Fugitive dust | 0 | 0 | 0 | 0 | 0 | 0 | − | − | 0 | 0 | 0 | 0 | 0 | − | − | − | − | − |

## 3 Results and discussion

### 3.1 Tempo-spatial variations of $PM_{2.5}$ in the PRD

The 4-month average $PM_{2.5}$ concentration for all six sites in the PRD was 37 μg/m³, which was slightly higher than the Grade II national standards for air quality (with an annual mean of 35 μg/m³). The chemical compositions of $PM_{2.5}$ in the PRD are shown in Fig. 2. OM had the highest contribution of 36.9%, suggesting severe organic pollution in the PRD. Other important components included $SO_4^{2-}$ (23.6%), $NH_4^+$ (10.9%), $NO_3^-$ (9.3%), EC (6.6%) and $Cl^-$ (0.9%). The major metallic components included K (1.5%), Na (1.1%), Fe (0.7%), Al (0.6%), and Ca (0.6%), and trace elements accounted for 1.0%. Fig. 3a shows the spatial distribution of the $PM_{2.5}$ and chemical components between six sites. The $PM_{2.5}$ pollution level in the PRD was distinctly higher in the northwestern hinterland (HS and MDS) and lower in the southern coastal areas (DM and DP). The DP background site had little local emission and was hardly influenced by the emissions from the PRD under both southerly flow and northerly flow. Thus, its air pollution reflects the large-scale regional air pollution. The average $PM_{2.5}$ concentration at DP was as high as 28 μg/m³, indicating that the PRD had a large amount of air pollution transported from outside this region. At the background DP site, the fractions of $Cl^-$ and $NO_3^-$ in $PM_{2.5}$ were the lowest of the six sites, i.e., 0.3% and 3.9%, respectively, suggesting that they had dominantly local sources in the PRD. The highest concentration level of $PM_{2.5}$ was observed at HS (suburban), which was influenced by the pollution transport of Foshan (industrial city) and Guangzhou (metropolis) under the northeastern wind, which is the most frequent wind in the PRD. The back trajectories of the air masses (Fig. S1) show that the northern monsoon prevails in winter and the southern monsoon prevails in summer in the PRD. Under the winter monsoon, the air masses mostly came from the inland and carried higher concentrations of air pollutants. However, under the summer monsoon, the air masses largely originated from the South China Sea and were clean. In addition, the frequent rainfall and higher planetary boundary layer (PBL) in summer in the PRD also favored the dispersion and removal of air pollutants (Huang et al., 2014b). Fig. 3b shows that the normalized seasonal variations of the major components in $PM_{2.5}$ in the PRD were evidently higher in winter and lower in summer, well consistent with the seasonal variations of monsoon and other meteorological factors as mentioned above.

Table 5 summarizes some previous studies that used similar filter-sampling and analytical methods to allow for a better comparison with this study. In 2002-2003, Hagler et al. (2006) also conducted observations and analysis of $PM_{2.5}$ in the PRD and Hong Kong region, nearly 12 years before this study, as shown in Table 5. Compared with Hagler's results, the $PM_{2.5}$ concentrations in this study decreased by 42% in Guangzhou (MDS) and 21% in Shenzhen (UT), especially OC, EC and $SO_4^{2-}$, which decreased significantly by 20%−47%, indicating that the measures taken to desulfurize coal-fired power plants, improve the fuel standards of motor vehicles and phase-out older and more polluting vehicles have played important roles in improving the air quality in the PRD region (People's Government of Guangdong Province, 2012). Compared with the $PM_{2.5}$ concentrations reported by other cities in China in recent years, the $PM_{2.5}$ concentrations in urban Guangzhou and Shenzhen in this study were 39%−63% lower than

those in Beijing (Huang et al., 2017) in northern China, Shanghai (Ming et al., 2017) in eastern China, and Chengdu (Wang et al., 2018) in western China. However, the $PM_{2.5}$ concentrations in urban Guangzhou and Shenzhen observed in this study were clearly higher than those in famous mega-cities in developed countries, such as Paris (Bressi et al., 2013), London (Rodríguez et al., 2007), and Los Angeles (Hasheminassab et al., 2014), while they were similar to those of Santiago (Villalobos et al., 2015) and Chuncheon (Cho et al., 2016). It should be highlighted that the higher concentration of $SO_4^{2-}$ in the urban atmosphere of the PRD is one of the major reasons leading to the higher degree of $PM_{2.5}$ pollution in the PRD compared to those in developed cities.

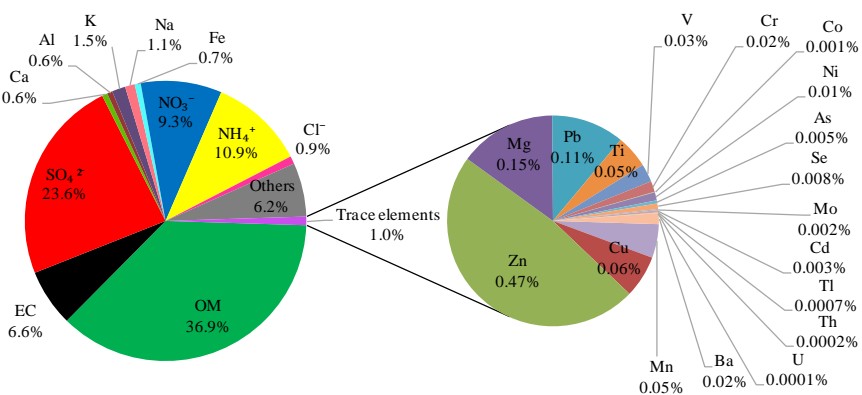

$PM_{2.5} = 37 \ \mu g/m^3$

**Fig. 2.** Chemical compositions of 4-month average $PM_{2.5}$ in the PRD region.

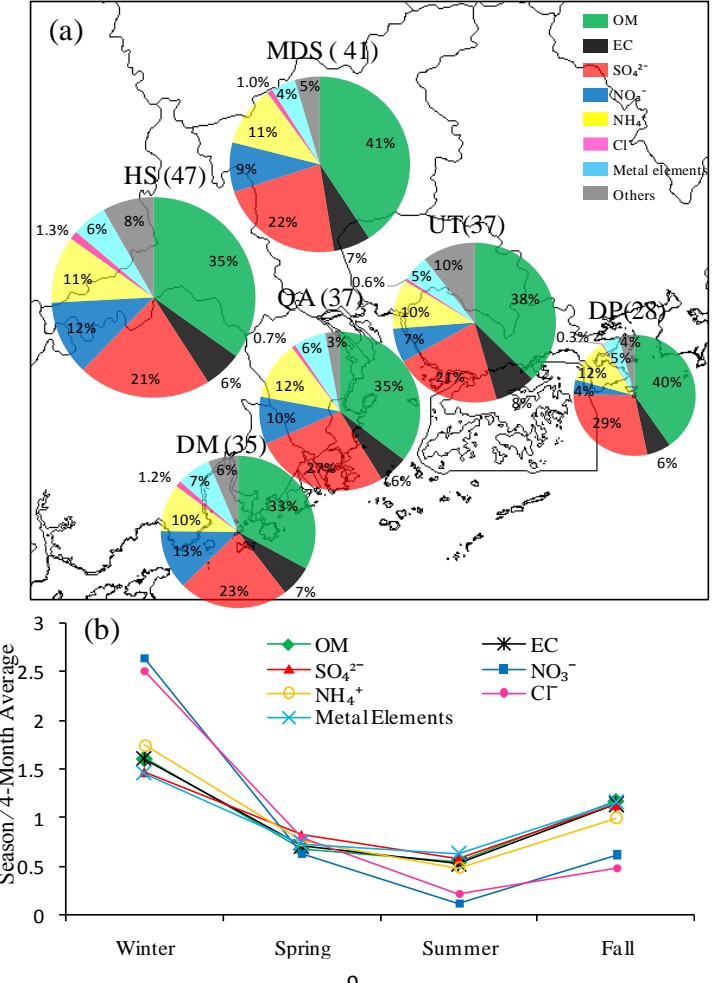

Fig. 3. The spatial distributions (a) and seasonal variations (b) of the PM$_{2.5}$ chemical compositions in the PRD. Sizes of the pie charts indicate the concentrations of PM$_{2.5}$ at the six sites, with the detailed numbers (unit: μg/m³) in brackets.

Table 5. The comparison of the major chemical compositions of PM$_{2.5}$ in typical cities (unit: μg/m³).

| Cities | Periods | PM$_{2.5}$ | OC | EC | SO$_4^{2-}$ | NO$_3^-$ | NH$_4^+$ | References |
|---|---|---|---|---|---|---|---|---|
| Zhuhai (DM) | 2015.1–2015.11 | 35 | 6.4 | 2.3 | 8.1 | 4.4 | 3.6 | This study |
| Zhuhai (QA) | | 37 | 7.2 | 2.2 | 9.9 | 3.5 | 4.4 | |
| Jiangmen (HS) | | 47 | 9.0 | 2.8 | 9.8 | 5.6 | 5.0 | |
| Guangzhou (MDS) | | 41 | 9.3 | 2.7 | 9.2 | 3.7 | 4.6 | |
| Shenzhen (UT) | | 37 | 7.8 | 3.0 | 8.0 | 2.6 | 3.7 | |
| Shenzhen (DP) | | 28 | 6.2 | 1.8 | 8.0 | 1.1 | 3.3 | |
| Hong Kong (Urban) | 2002.10–2003.6 | 34.3 | 6.6 | 1.9 | 9.3 | 1.0 | 2.5 | Hagler et al., 2006 |
| Shenzhen (Urban) | | 47.1 | 11.1 | 3.9 | 10.0 | 2.3 | 3.2 | |
| Guangzhou (Urban) | | 70.6 | 17.6 | 4.4 | 14.7 | 4.0 | 4.5 | |
| Beijing | 2014.6–2015.4 | 99.5 | 15.5 | 6.2 | 14.3 | 17.9 | 11.5 | Huang et al., 2017 |
| Shanghai | 2013.9–2014.8 | 94.6 | 9.89 | 1.63 | 14.5 | 18.0 | 8.13 | Ming et al., 2017 |
| Chengdu/Sichuan | 2014.10–2015.7 | 67.0 | 10.9 | 3.6 | 11.2 | 9.1 | 7.2 | Wang et al., 2018 |
| Paris/France | 2009.9–2010.9 | 14.8 | 3.0 | 1.4 | 2.0 | 2.9 | 1.4 | Bressi et al., 2013 |
| London/United Kingdom | 2003.12–2005.4 | 31.0 | 5.6 | 1.6 | 2.8 | 3.5 | 2.1 | Rodríguez et al., 2007 |
| Los Angeles/United States | 2002–2013 | 17.1 | 2.2 | 1.3 | 2.7 | 4.9 | 0.1 | Hasheminassab et al., 2014 |
| Santiago/Chile | 2013.3–2013.10 | 40 | 12.1 | 4.3 | 1.9 | 7.1 | 3.3 | Villalobos et al., 2015 |
| Chuncheon/Korea | 2013.1–214.12 | 34.6 | 9.0 | 1.6 | 3.9 | 2.8 | 2.0 | Cho et al., 2016 |

## 3.2 Source apportionment of PM$_{2.5}$ using ME-2

The solutions of 9–12 factors of the ME-2 were modeled with the four factors constrained in Table 4, using the SoFi tool, an implementation of ME-2 (Canonaco et al., 2013). Again, the nine-factor solution provided the most reasonable source profiles, since non-interpretable factors were produced (e.g., a Ti-high factor) when more factors were set to run ME-2. Based on the EV and the contributed concentrations of species in each factor shown in Fig. 4, the sources of PM$_{2.5}$ can be judged as follows: (1) the first factor was explained as secondary sulfate-rich, which had large EV values of SO$_4^{2-}$ and NH$_4^+$. (2) The second factor was explained as secondary nitrate-rich, which had significant EV values of NO$_3^-$ and NH$_4^+$. (3) The third factor was related to sea salt due to the large EV values and concentrations of Na and Mg. However, the low Cl$^-$ concentration and high SO$_4^{2-}$ concentration implied that SO$_4^{2-}$ replaced Cl$^-$ during the sea salt aging process. Therefore, this factor was identified as aged sea salt (Yuan et al., 2006a). (4) The fourth factor was identified as fugitive dust due to its significant EV values of Al, Ca, Mg and Fe. In this study, the undetermined mass of O and Si in this factor was compensated using the elemental abundance in dust particles in Table S1 (Taylor and Mclennan, 1995). (5) The fifth factor was identified as biomass burning due to its significant characteristic value of K (Yamasoe et al., 2000). (6) The sixth factor had high concentrations and large EV values of OM and EC, as well as a certain range of EV values of Fe and Zn, which were related to tires and the brake wear of motor vehicles (Yuan et al., 2006a; He et al., 2011). Therefore, this factor was identified as vehicle emissions. (7) The

seventh factor had a high EV value of $Cl^-$ and certain concentrations of OM, EC, $SO_4^{2-}$ and $NO_3^-$,
implying a combustion source. This factor was identified as coal burning, which was a major
source of $Cl^-$ in the PRD (Wang et al., 2015). (8) The eighth factor had large EV values of Zn, Cd
and Pb, and certain concentrations of OM and EC. Zn, Cd and Pb had high enrichment factors
(Table S2) of 821, 4121 and 663, respectively, and were thus considered to be related to industrial
emissions (Wang et al., 2015). (9) The last factor had large EV values of V and Ni. V and Ni were
predominantly derived from heavy oil combustion, and they had high enrichment factors (Table
S2) of 64 and 89, respectively. Heavy oil was related to ship emissions in the PRD (Chow et al.,
2002; Huang et al., 2014b). Although these nine factors of the ME-2 modeling generally showed
high correlations ($R^2$=0.81–0.97) with the corresponding factors of the PMF modeling in terms of
time series, it is easy to see that the ME-2 modeling provided a better $Q_{true}/Q_{exp}$ ratio (1.2) than
that of the PMF modeling ($Q_{true}/Q_{exp}$=2.5), indicating that the species residuals were decreased in
the ME-2 modeling, and the EV values of tracers (e.g., $SO_4^{2-}$, $NO_3^-$, OM, EC, $Cl^-$, V, Ni, Pb and
Cd) were assigned to factors more intensively. Therefore, it is concluded that the source
apportionment results of the ME-2 modeling were more environmentally meaningful and
statistically better than those of the PMF modeling.
In this study, secondary organic aerosol (SOA) did not appear as a single factor, even if we
run the ME-2 with ten or more factors. SOA can usually be described by low-volatile oxygenated
organic aerosol (LV-OOA) and semi-volatile oxygenated organic aerosol (SV-OOA), based on the
volatility and oxidation state of organics (Jimenez et al., 2009). In previous studies (e.g., He et al.,
2011; Lanz et al., 2007; Ulbrich et al., 2009), the time series of LV-OOA and SV-OOA were
highly correlated with those of sulfate and nitrate, respectively, implying that LV-OOA and
sulfate (or SV-OOA and nitrate) cannot be separated easily in cluster analysis, especially when
there is no effective tracer of SOA. In this study, the high OM concentration in the secondary
sulfate-rich factor was considered to represent LV-OOA, while the high OM concentration in the
secondary nitrate-rich factor was considered to represent SV-OOA (Yuan et al., 2006b; He et al.,
2011). Therefore, it should be acknowledged that mixed secondary factors cannot be solved even
using ME-2. In this study, however, an SOA factor can be reasonably extracted from the
secondary sulfate-rich and secondary nitrate-rich factors and regarded as the sum of the OM
concentrations in these two factors, i.e., LV-OOA+SV-OOA, leaving the remaining mass as
independent secondary sulfate and secondary nitrate.

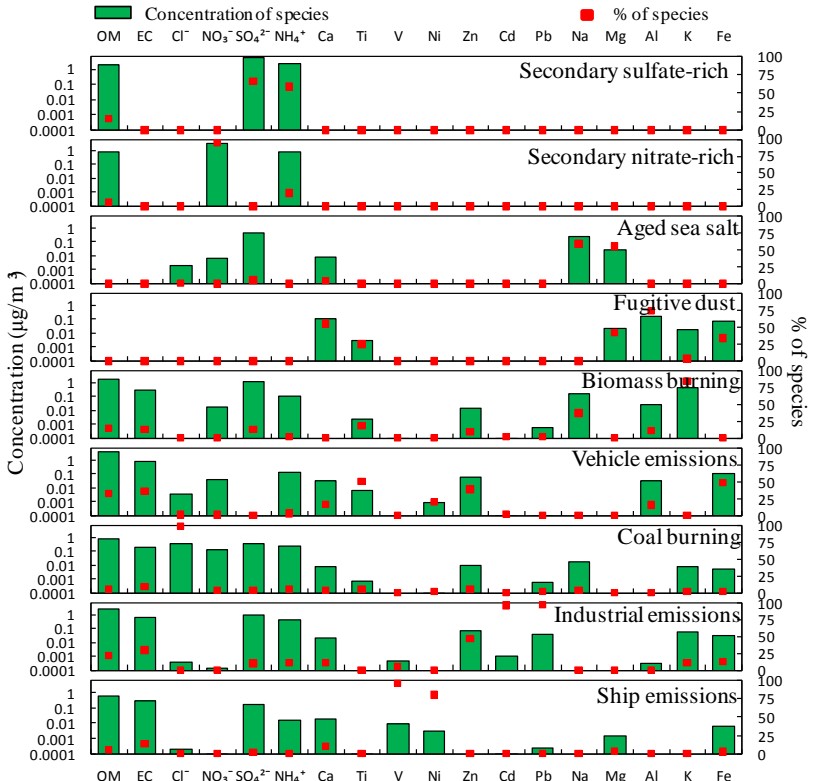

**Fig. 4.** The factor profiles and explained variations of the ME-2 modeling.

Fig. 5 shows the 4-month average contributions of the PM$_{2.5}$ sources in the PRD in 2015 based on the source apportionment of ME-2. The total secondary aerosols accounted for 39% of PM$_{2.5}$ in the PRD, which were secondary sulfate (21%), secondary nitrate (11%) and SOA (7%). However, the identified primary particulates contributed 54% of PM$_{2.5}$, which comprised vehicle emissions (14%), industrial emissions (13%), biomass burning (11%), coal burning (6%), fugitive dust (5%), ship emissions (3%) and aged sea salt (2%). The unidentified sources, including both the residual from ME-2 and the unmeasured species, accounted for 7%.

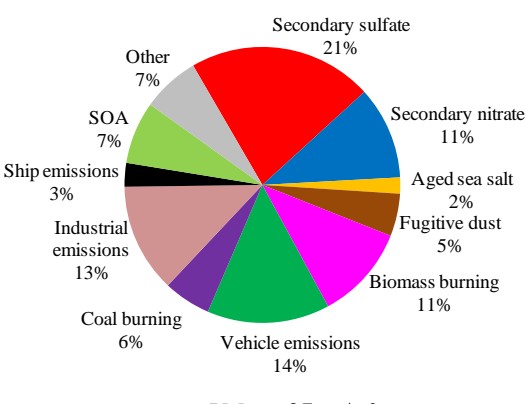

**Fig. 5.** The 4-month average contributions of PM$_{2.5}$ sources in the PRD.

**3.3 Tempo-spatial variations of sources in the PRD**

The spatial distributions of the PM$_{2.5}$ sources between six sites are shown in Fig. 6a. Secondary sulfate represented the largest fraction (31%) of PM$_{2.5}$ at DP, indicating that it was a major air pollutant in the air mass transported to the PRD. Vehicle emissions also contributed

relatively highly to urban sites (18% in MDS and 17% in UT). Industrial emissions, biomass
burning, secondary nitrate, and coal burning contributed larger fractions of $PM_{2.5}$ at HS, which
could be attributed to both strong local sources (e.g., the surrounding township factories and
farmlands) and regional transport from upwind cities at this site. Fugitive dust, which is primarily
related to construction activities, was relatively high at DM (9%). The contributions of ship
emissions and aged sea salt were the highest at QA due to its being located on Qi-Ao Island in the
Pearl River Estuary, which records the greatest impact from the sea. SOA contributed similar
amounts (7%−8%) at all sites. It should be noted that, although QA was a background site without
local anthropogenic sources, its $PM_{2.5}$ level was moderate in the PRD, indicating that QA was
impacted by severe regional transport from the surrounding cities.
Fig. 6b shows the seasonal variations of the major sources of $PM_{2.5}$ in the PRD. The
contributions of most sources were higher in winter and lower in summer, e.g., secondary sulfate,
secondary nitrate, fugitive dust, biomass burning, vehicle emissions, coal burning, industrial
emissions and SOA; these sources were greatly influenced by the seasonal variations of monsoon,
rainfall and PBL, as discussed in Section 3.1. For example, although secondary sulfate was proven
to be a typical regional pollutant in the PRD (Huang et al., 2014b; Zou et al., 2017), the more
polluted continental air mass in the winter monsoon made its concentrations in winter much higher
than in summer. The semi-volatile secondary ammonium nitrate was also significantly affected by
seasonal ambient temperatures. In contrast, the average contributions of aged sea salt and ship
emissions for the whole region displayed little seasonal variations, consistent with that the
emissions were from local surrounding sea areas.
Previous studies of the source apportionment of bulk $PM_{2.5}$ in the PRD have mainly focused
on Guangzhou, Dongguan and Shenzhen, as seen in Table 6. It can be seen that in those studies,
$PM_{2.5}$ was apportioned to 6–9 sources and that secondary sulfate was the prominent source,
although the results of different studies exhibited certain differences due to the use of different
models or data inputs. Compared with the study of Huang et al. (2014b) in Shenzhen in 2009, the
contributions of secondary sulfate and vehicle emissions in Shenzhen in this study were obviously
lower due to power plant desulfurization and motor vehicle oil upgrades in recent years (People's
Government of Shenzhen Municipality, 2013). Compared with previous studies in Guangzhou,
this study attained more $PM_{2.5}$ sources, which can more clearly describe the source structure of
$PM_{2.5}$ in this region, especially industrial emissions (11%). The PRD region has experienced a
high degree of industrialization; thus, industrial sources should be a major source, contributing 8.1%
of $PM_{2.5}$ reported by the Guangzhou Environmental Protection Bureau (2017), similar to our
results. Tao et al. (2017) apportioned $PM_{2.5}$ to 6 sources using PMF in Guangzhou, including some
mixed sources. For example, ship emissions in Tao's study may not actually represent a primary
source due to the significant existence of some secondary inorganics and sea salt in the source
profile; thus, they obtained a significantly higher contribution (17%) than that in our study. Ship
emissions were unidentified in Huang's study (2014a) in Guangzhou.

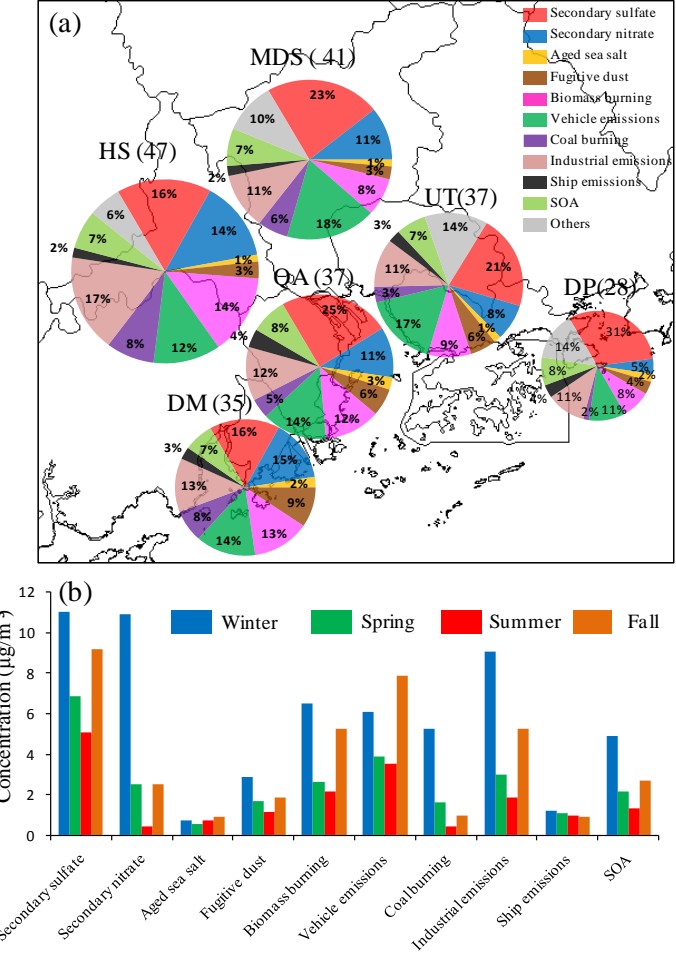


**Fig. 6.** The spatial distributions (a) and seasonal variations (b) of PM$_{2.5}$ sources in the PRD. Sizes of the pie charts indicate the concentrations of PM$_{2.5}$ at the six sites, with the detailed numbers (unit: μg/m³) in brackets.


**Table 6.** Comparison of the results of source apportionment of PM$_{2.5}$ in the PRD.

| Cities | Periods | Model | Results | References |
|---|---|---|---|---|
| Shenzhen | 2015.1—2015.11 | ME-2 | Secondary sulfate (21%), secondary nitrate (8%) and SOA (7%), vehicle emissions (17%), industrial emissions (11%), biomass burning (9%), coal burning (3%), fugitive dust (6%), ship emissions (3%) and aged sea salt (1%). | This study |
| Shenzhen | 2009.1—2009.12 | PMF | Secondary sulfate (30.0%), vehicular emission (26.9%), biomass burning (9.8), secondary nitrate (9.3%), high chloride (3.8%), heavy oil combustion (3.6%), sea salt (2.6%), dust (2.5%), metallurgical industry (2.1%). | Huang et al. (2014b) |
| Guangzhou | 2015.1—2015.11 | ME-2 | Secondary sulfate (23%), secondary nitrate (11%), SOA (7%), vehicle emissions (18%), industrial emissions (11%), biomass burning (8%), coal burning (6%), fugitive dust (3%), ship emissions (2%) and aged sea salt (1%). | This study |
| Guangzhou | 2014.1—2014.12 | PMF | Secondary sulfate and biomass burning (38%), ship emissions (17%), coal combustion (15%), traffic emissions (10%), secondary nitrate and chloride (12%), soil dust (7%). | Tao et al. (2017) |
| Guangzhou | 2015.1—2015.2 | ME-2 | Secondary sulfate (20%), secondary nitrate (16%), SOA (8%), vehicle emissions (11%), industrial emissions (13%), biomass burning (6%), coal burning (9%), fugitive dust (2%), ship emissions (1%) and aged sea salt (1%). | This study |

| Guangzhou | 2013.1 | ME-2 | Secondary inorganic-rich (59.0%), secondary organic-rich (18.1%), traffic (8.6%), coal burning (3.4%), biomass burning (6.7%), cooking (0.8%), dust related (3.4%). | Huang et al. (2014a) |
|---|---|---|---|---|
| Dongguan | 2013.12—2014.11 | PMF | Secondary sulfate (20%), secondary nitrate (8%), SOA (10%), vehicle emissions (21%), industrial emissions (7%), biomass burning (11%), coal burning (5%), fugitive dust (8%), ship emissions (6%). | Zou et al. (2017) |
| Dongguan | 2010.2—2012.12 | PMF | Secondary sulfate (27%), secondary nitrate (19%), industrial emission (15%), biomass burning (9%) and coal combustion (9%); ship emissions/sea salt, vehicle exhaust, plastic burning and dust no more than 7%. | Wang et al. (2015) |


**3.4 Identification of high-emission areas in the PRD in typical meteorological conditions**

Fig. 7 shows the contributions of PM$_{2.5}$ sources under southerly flow and northerly flow conditions in the PRD, based on the classification of weather types in Section 2.2. Southerly flow primarily originated from the South China Sea and carried clean ocean air masses to the PRD with overall PM$_{2.5}$ values of 15 μg/m³. As shown in Fig. 7, secondary sulfate (19%), vehicle emissions (15%) and biomass burning (11%) had higher contributions under southerly flow. In contrast, in northerly flow, the level of PM$_{2.5}$ (82 μg/m³) was 4.5 times higher than that of southerly flow due to the transport of polluted air masses southward from the north mainland. Under northerly flow, secondary sulfate (18%) and biomass burning (10%) were still the major sources, but secondary nitrate became the dominant source of PM$_{2.5}$, accounting for 20% of PM$_{2.5}$. In addition, industrial emissions also exhibited a relatively high contribution (14%).

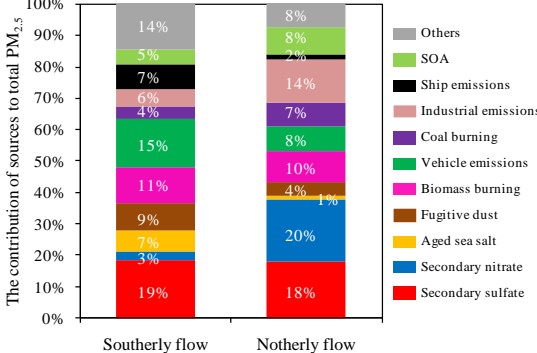

**Fig. 7.** The contributions of PM$_{2.5}$ sources under southerly flow and northerly flow conditions in the PRD.

The spatial distributions of the PM$_{2.5}$ sources under southerly flow and northerly flow are shown in Fig. 8. The high-emission areas for different sources identified by the discussion below are marked on the map in Fig. 9. The average concentration levels of aged sea salt were similar in the summer southerly flow and the winter northerly flow, reflecting local release of sea salt. The spatial distribution of aged sea salt among the different sites was a complex result of the site locations relative to the sea and meteorological conditions, e.g., wind and tide. A relatively high level of aged sea salt was observed at the Qi-Ao Island (QA), especially in the northerly flow, which can be attributed to that the QA site was surrounded by the sea and had lower wind speeds in the northerly flow (in Table 3).

The influences of ship emissions exhibited large differences between six sites, showing significant local characteristics. In addition, the ship emissions have similar average concentrations in the summer southerly flow and winter northerly flow, also reflecting the emissions of local ports in the PRD region. The concentrations of ship emissions were the highest

at DP under southerly flow, mainly due to the impact of vessels in the upwind Yiantian Port, while
they were the highest at QA under northerly flow, primarily due to the effects of the upwind
Nansha Port, as shown in Fig. 9. Yantian Port and Nansha Port are among the ten largest ports in
the world (Hong Kong Marine Department, 2012).
The contributions of fugitive dust also exhibited significant differences between six sites,
which are consistent with local construction activities. DM is located in a newly developed zone
that has experienced relatively high levels of fugitive dust during southerly flow and northerly
flow due to active construction activities. Sample records indicate that the high value of fugitive
dust at UT under southerly flow maybe related to its surrounding short-term road construction
project, while the high value at QA under northerly flow maybe related to the reconstruction
project of the adjacent Nansha Port (Guangzhou Municipal People's Government, 2015).
Motor vehicles are a common source of air pollution in the highly urbanized and
industrialized PRD region. The average concentration of vehicle emissions during northerly flow
was nearly 3-fold that during southerly flow. Under southerly flow, MDS, HS and UT, which are
located in the hinterland of the PRD, had much higher levels of vehicle emissions than the other
three sites; in particular, the highest level at the urban MDS site was caused by the high density of
motor vehicles in Guangzhou. Under northerly flow, the highest concentration of vehicle
emissions was still at the urban MDS site, while QA also recorded the prominent contribution of
vehicle emissions, which was probably closely related to the container trucks in the neighboring
Nansha Port. It should be noted that the concentration of vehicle emissions at the background DP
site exceeded half the regional average value, approaching 4 $\mu g/m^3$, thus indicating that vehicle
emissions had a significant impact on the regional transport of air masses from the north.
During southerly air flow, the background DP and QA sites and the urban UT site all
recorded similar concentrations of secondary sulfate, suggesting that the secondary sulfate at these
sites was dominated by regional transport from the southern ocean with heavy vessel transport and
had little to do with the urban emissions at UT. Kuang et al. (2015) also found that ship emissions
could be a major source of secondary sulfate in the PRD in summer. HS and MDS had
significantly higher concentrations than their upwind site, DM, suggesting that the area between
MDS and HS could be a high-$SO_2$-emission area, which is consistent with the fact that this area is
an intensive industrial area. During northerly air flow in winter, HS and DM had lower
concentrations than the four upwind sites, i.e., MDS, QA, UT, and especially DP (the background
site), indicating that secondary sulfate could mainly be derived from regional transport from
outside the PRD in this season. Although the industrial area between HS and MDS could emit
significant amounts of $SO_2$, the lower temperatures and dry air in winter did not appear to favor
the quick conversion of $SO_2$ to secondary sulfate. Since both secondary sulfate and LV-OOA
belong to a mixed factor with fixed proportions, the spatial distribution of secondary sulfate also
reflects the corresponding characteristics of LV-OOA.
The spatial distributions of coal burning were significantly different between the six sites
during periods of both south wind and north wind, thus showing conspicuous local characteristics.
The contribution of coal burning was higher at MDS under southerly flow and higher at HS under
northerly flow. Most of the coals in the PRD were consumed by thermal power plants, but there
were no coal-fired power plants near the urban MDS and background DP sites. Therefore, it is
speculated that the high-emission areas of coal burning sources mainly exist in the region between
HS and MDS, as shown in Fig. 9. The distributions of coal-fired power plants in Guangdong
(Wang et al. 2017) reveal that some important coal-fired power plants are distributed in this region.
Additionally, DM also exhibited relatively obvious contributions of coal burning during southerly
flow and northerly flow, which is also consistent with the distribution of coal-fired power plants in
the vicinity.

The average concentration of secondary nitrate during northerly flow in winter was 40 times
greater than that during southerly flow in summer; this occurred not only because of the
unfavorable conditions of atmospheric diffusion in winter but also due to the high semi-volatility
of ammonium nitrate, which cannot stably exist in fine particles in the PRD during hot summer
(Huang et al. 2006). Under southerly flow conditions, the concentrations of secondary nitrate
presented prominent differences between six sites, showing local characteristics. Moreover, the
relatively low concentrations at the background DP site during northerly flow also indicated that
secondary nitrate mainly originated from the interior of the PRD. The spatial distribution
characteristics of secondary nitrate were very similar to those of coal burning, with the highest
occurring at MDS under southerly flow, the highest occurring at HS under northerly flow and
significantly high values occurring at DM under southerly and northerly flow, displaying that the
NOx emissions produced by coal burning maybe the main reason for the high nitrate levels in
those areas. Since both secondary nitrate and SV-OOA belong to a mixed factor with fixed
proportions, the spatial distribution of secondary nitrate also reflects the corresponding
characteristics of SV-OOA.

Under southerly flow, the influence of industrial emissions differed vastly between six sites,
showing obvious local characteristics. Under northerly flow, the average concentration of
industrial emissions reached 14-fold that of southerly flow, and the high contributions at
background DP suggested that regional transport probably dominated the industrial sources of fine
particulate matter in the PRD in winter. HS had the highest concentration of industrial emissions
during southerly flow and northerly flow conditions, which is consistent with the dense factories
present in the surrounding area (Hu, 2004; Environmental Protection Agency of Jiangmen City,
2017). In addition, the contribution of industrial emissions was relatively high at MDS during
southerly flow and relatively high at QA during northerly flow, which supports the inference that a
high-emission region of industrial sources was located between MDS and QA, as seen in Fig. 9.

The impacts of biomass burning exhibited relatively large differences between six sites
during both south and north wind conditions, presenting somewhat local characteristics. Suburban
HS site had relatively high biomass burning levels during southerly flow and northerly flow,
which should be related to the presence of many farmlands in its vicinity and thus the popular
events of open burning and residential burning of biomass wastes. The concentrations of biomass
burning were relatively high at the urban MDS site during southerly flow and relatively high at the
background QA site during northerly flow, implying that there was a high-emission area of
biomass burning between MDS and QA, as shown in Fig. 9. Those spatial distribution
characteristics of biomass burning were similar to those of industrial emissions in the PRD,
suggesting that not only the combustion of residential biomass but also the use of industrial
biomass-boilers could make important contributions to $PM_{2.5}$ in the PRD.

As a summary, the central PRD area, i.e., the middle region between MDS, HS and QA (the
shaded region in Fig. 9), represents the most important pollutant emissions area in the PRD; these
emissions include $SO_2$, NOx, coal burning, biomass burning, industrial emissions and vehicle
emissions, thus leading to high pollution levels in the PRD. Therefore, this area is a key area for
pollution control in the PRD. Primary fine particulate matter and SO₂ from ship emissions had
significant impacts on PM₂.₅ in the southern coastal area of the PRD during summer southerly
flow, and special attention must be paid to them.

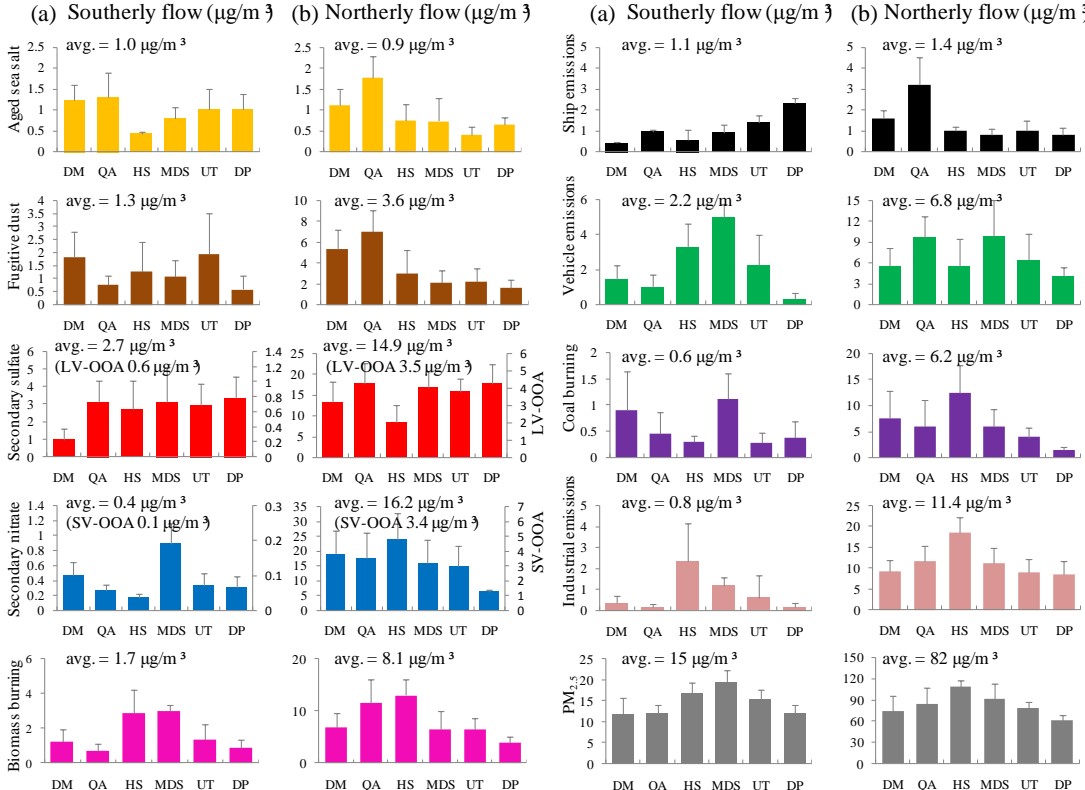


**Fig. 8.** The average contributions of PM₂.₅ sources at six sites in the PRD: (a) those in southerly flow, (b) those in
northerly flow.

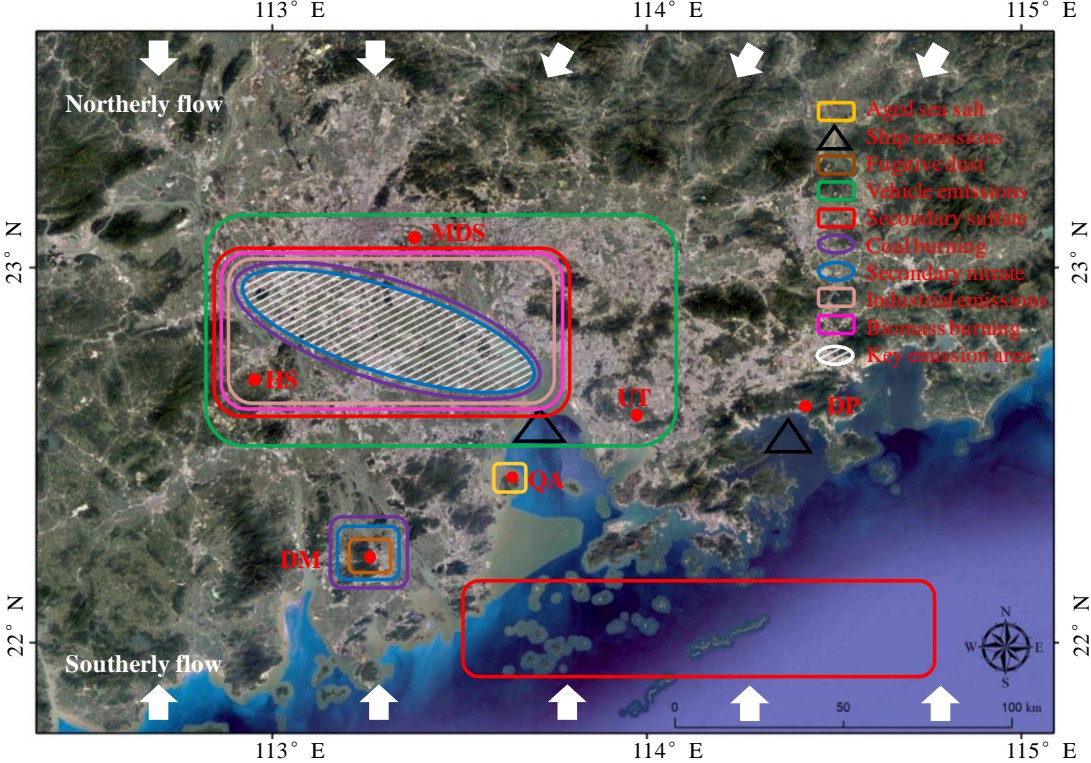


**Fig. 9.** The schematic diagram of high-emission areas in the PRD (map from Google Earth). The white shaded area
indicates the key emission area for the multiple sources of $SO_2$, NOx, coal burning, biomass burning, industrial
emissions and vehicle emissions, and is explained further in the text.
**3.5 Distinguishing local and regional PM$_{2.5}$ pollution in the PRD**

The analyses presented in Section 3.4 indicate that the secondary sulfates at the four southern

coastal sites (DM, QA, UT and DP) in the PRD were almost entirely derived from the conversion
of $SO_2$ from the emissions of ships in the southern ocean during southerly flow, contributing
approximately 20% of the average PM$_{2.5}$ (13 μg/m³) at the four sites. Considering that the ship
emissions directly contributed approximately 10% of the average PM$_{2.5}$ at the four sites, the total
ship emissions contributed approximately 30% of PM$_{2.5}$ in the southern coastal PRD area and
acted as the largest source of PM$_{2.5}$. Under northerly flow conditions, the background DP site,
which was barely affected by pollution emissions within the PRD, reflected regional transport
from the north air mass outside the PRD, while the background QA site reflected the superposition
effect of regional background pollution and the input of the most serious pollution area in the PRD.
The consistency of the secondary sulfate concentrations at the background QA and DP sites was
interpreted to reflect almost the same regional background effect during northerly flow; thus, the
differences in the six anthropogenic sources between the two background sites, including
secondary nitrate (and SV-OOA), biomass burning, industrial emissions, coal burning, vehicle
emissions and ship emissions, could be used to trace the internal inputs from the most serious
pollution area within the PRD to the downwind area. The internal inputs of six anthropogenic
sources to the corresponding sources of PM$_{2.5}$ at the background QA site were 66%, 67%, 28%,
76%, 59% and 75%, respectively, and the total internal input of 37.7 μg/m³ accounted for 45% of
PM$_{2.5}$ at the background QA site (83 μg/m³), showing that the local contributions of anthropogenic
pollution emissions in the key source area of the PRD were still crucial in winter but lower than
the contribution of the regional background. Ignoring natural sources, such as aged sea salt and
fugitive dust, under northerly flow, the contributions of other anthropogenic sources to DP were
considered to represent regional background pollution (47.5 μg/m³), and the differences in their
corresponding source concentrations between QA and DP were expected to represent the local
emissions of source areas in the PRD. Therefore, the source structures in the regional background
air mass and local emissions of heavy pollution sources area in the PRD are shown in Fig. 10.
Secondary sulfate and LV-OOA occupied the vast majority (45.6%) of the regional background air
mass from the northern mainland, followed by industrial emissions (17.8%), secondary nitrate and
SV-OOA (15.5%). However, the major sources between the sources output by local emissions
from the heavy pollution source area of the PRD were secondary nitrate and SV-OOA (37.3%),
biomass burning (20.6%), vehicle emissions (14.9%) and coal burning (11.9%). Therefore,
measures implemented for the effective control of PM$_{2.5}$ in the PRD should focus on local controls
and regional joint prevention and control under winter northerly flow conditions.

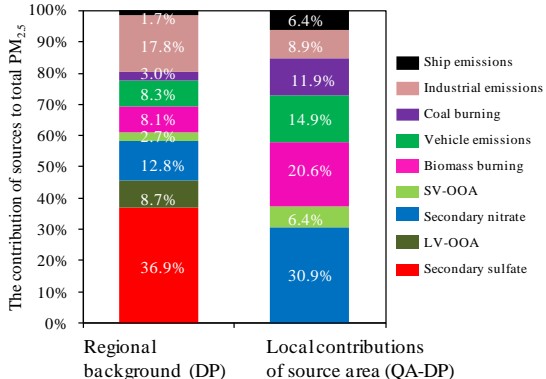


**Fig. 10.** The PM$_{2.5}$ source structures in regional background air and local contributions of the central PRD area
under northerly flow.

**4 Conclusions**
The PRD is one of the largest agglomeration of cities in the world, and its air quality has
largely improved in the past ten years. To reveal the current PM$_{2.5}$ pollution characteristics on a
regional scale in the PRD, six sampling sites were selected to conduct 4 months of sampling and
chemical analysis in 2015; then, the source exploration of PM$_{2.5}$ was performed using a novel
method. The conclusions are described below.
(1) The 4-month average PM$_{2.5}$ concentration for all six sites in the PRD was 37 μg/m³, of which
OM, $SO_4^{2-}$, $NH_4^+$, $NO_3^-$, EC, metal elements and $Cl^-$ contributed 36.9%, 23.6%, 10.9%, 9.3%,
6.6%, 6.5% and 0.9%, respectively. The tempo-spatial PM$_{2.5}$ variations were generally
characterized as being higher in the north inland region and higher in winter.
(2) This study revealed that the ME-2 model produced more environmentally meaningful and
statistically robust results of source apportionment than the traditional PMF model. Secondary
sulfate was found to be the dominant source of PM$_{2.5}$ in the PRD, at 21%, followed by vehicle
emissions (14%), industrial emissions (13%), secondary nitrate (11%), biomass burning (11%),
SOA (7%), coal burning (6%), fugitive dust (5%), ship emissions (3%) and aged sea salt (2%).
Only aged sea salt and ship emissions did not show obvious seasonal variations.
(3) Based on the spatial distribution characteristics of PM$_{2.5}$ sources under typical southerly and
northerly airflow conditions, the central PRD area between MDS, HS and QA is identified as a
key area for source emissions, including $SO_2$, NOx, coal burning, biomass burning, industrial
emissions and vehicle emissions, and thus deserves more attention when implementing local
pollution control in the PRD. In addition, ship emissions should be controlled more strictly during
summer due to its contribution of approximately 30% of PM$_{2.5}$ in the southern coastal area of the
PRD under southerly air flow.
(4) Under typical winter northerly flow, the contributions of anthropogenic pollution emissions in
the central PRD area contributed 37.7 μg/m³ (45% of PM$_{2.5}$) to the regional background air.
Secondary sulfate (36.9%), industrial emissions (17.8%), and secondary nitrate SV-OOA (12.8%)
were the major PM$_{2.5}$ sources for the PM$_{2.5}$ transported in the regional background air mass, while
secondary nitrate (30.9%), biomass burning (20.6%), vehicle emissions (14.9%) and coal burning
(11.9%) were the major sources for the PM$_{2.5}$ produced in the central PRD area. Therefore,
effective control measures of PM$_{2.5}$ in the PRD in the future should pay more attention to both
local controls and regional joint prevention.

**Acknowledgments**

This work was supported by the National Natural Science Foundation of China (91744202; 41622304) and the Science and Technology Plan of Shenzhen Municipality (JCYJ20170412150626172, JCYJ20170306164713148).

615

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
