# Peer review of "Exploration of PM2.5 sources on the regional scale in the"

_Atmospheric Chemistry and Physics, 2018_

## Referee Comment (RC1) · Anonymous Referee #1 · 14 Apr 2018

General comments:

Huang et al. investigate the sources of PM2.5 in the Pearl River Delta (PRD) region of China, determining whether the sources are local or regional and how they vary under different meteorological conditions based on six sites representing urban, suburban, and background locations. The authors present detailed chemical composition results from data collected at each of the sites for approximately one month during each the main four seasons to represent the variability during a full calendar year. Both PMF and ME-2 were applied to the data to identify potential sources of PM2.5 in the area, which were subsequently correlated with meteorological conditions, such as monsoons, to

further identify the importance of each of the sources including local versus regional nature and temporal significance. The authors compared the findings from this work with previous studies in the same areas as well as putting the results into a global context. Ultimately, the authors were able to identify key emission sources and locations that should be targeted in future pollution control measures.

Although the scientific quality of the work is good, the authors do not obviously highlight the uniqueness of this study. The data presented are new thus add to the scientific knowledge and understanding of the PRD and the methods used, particularly ME-2, appear to be novel in that they are applied to a unique dataset. If this is the case, the authors should include a sentence or two in the appropriate places within the manuscript (e.g. abstract). The scientific methods and assumptions are valid and the results are generally sufficient to support the interpretations and conclusions although some additional evidence or explanation is needed (see specific comments). Numerous pie charts are presented in the manuscript; the authors should consider moving some of these to the supplementary material to reduce the length of the manuscript or use a different style plot to distinguish between the different types of results being presented. The figures currently in the supplementary material need to be greatly improved in terms of clarity of the images as well as the addition of legends where possible. The manuscript generally flowed well but it could do with some slight reordering, especially the section describing the meteorological conditions, to make the manuscript flow even better.

Despite there being some major points that need to be revised, the overall quality of the work presented and manuscript itself are good; an interesting and enjoyable read. Once the revisions above and the comments below are addressed, I recommend this manuscript be published in ACP.

Specific comments:

Abstract, line 21: It is stated that the 'regional annual average PM2.5 concentration was

determined'. This is misleading as there were only ∼4 months of measurements spanning the year, with samples taken every other day. Although those four months may be representative of the main pollution conditions, it should be mentioned or clarified that a full year of data was not obtained to determine/estimate this annual average.

Introduction, lines 50-51: Why are these noteworthy provinces? Please consider adding a few words as to why these are being highlighted.

Introduction, line 62: It is stated that the previous studies in the PRD provinces 'lacked the extensive representation of the PRD'. Please qualify this statement e.g. is it because only single locations were studied and that is the uniqueness of this study as several locations are studied at the same time.

Introduction, lines 65-67: Despite some of PMF's limitations, it is the first step for the application of ME-2 to a dataset. Further, as PMF does not require a prior information, new sources could be identified as a result both in terms of newly identified as a source in a given location or a newly identified emission source overall. Please add a sentence or two to acknowledge that PMF is usually the first step in factor analysis using ME-2, especially as the a priori information used for running ME-2 typically uses the factor profiles identified from PMF and/or identifies a number of factors that should be considered when running ME-2.

Introduction, lines 69-74: As mentioned, organic aerosols have been successfully apportioned using ME-2 via SoFi. As this study uses both organic and inorganic species it would be good to point this out as being novel. If it is the first study of its kind to apply the model to this dataset (in terms of the species and/or measurement period and location) then this should be highlighted in the manuscript in the appropriate places such as the abstract and later in the introduction. If this is not a unique case then something along the lines above should be mentioned in any case along with a citation of similar cases for comparison.

Section 2.1, lines 93-94: The current way in which the sampling periods are described

are misleading as 'January-February' could be interpreted as being two full months whereas in fact it is a period of one month spanning two months. Add a few words clarifying that each sampling period for the seasons is one month and refer to table 2, where the exact sampling dates are noted.

Section 2.1, lines 100-101: 'two different types of samplers sampled' – clarify that it is the two samplers that were used in this study that were compared. The results of the inter-instrument comparison 'yielded a relative deviation of less than 5% for PM2.5 mass concentrations'. How many samples were obtained for this comparison? How was the 5% calculated/determined? Please consider adding something to the manuscript on this.

Section 2.1, in general: There is no mention of the exact number of samples that were obtained and whether there were any issues with any of them. Are all ∼15 samples from each season valid and run as intended? What QA/QC was performed on the samples (standard laboratory QC and overall QA)?

Section 2.2, lines 146-147: Please provide examples of the 'sources of uncertainty that contributed little to the total uncertainty'.

Section 2.2, lines 159-160: Please expand on why a factor of 2 was applied to the estimated uncertainties. Specifically, please explain why a factor of 2 was chosen. If this is this a typical factor to apply, please provide a reference.

Section 2.3, line 168: It is not typical to refer to later sections in a manuscript. Perhaps consider summarizing what is in the later sections here or rephrase this sentence so that Section 3.2 can be referenced but the reader does not have to read that section at this point.

Section 3.1., lines 211-215: Figure 3b does not show that the seasonal variations in the major PM2.5 components were correlated with monsoon characteristics. Please expand on this, clarify, and/or provide additional evidence for this statement. Similarly,

please expand on/clarify how figure S1 shows that the northern monsoon prevails in winter and the southern monsoon prevails in summer.

Section 3.2, lines 251-257: Please comment on why it might not have been possibly to separate the secondary sulfate and LV-OOA as two separate factors as well as SV-OOA and secondary nitrate. Having a mixed factor is something the authors note as being a downfall of the PMF results so it needs to be acknowledged that even with SoFi there is a mixed factor. To confirm LV-OOA/OOA-1 and SV-OOA/OOA-2 factors in other studies, the time series is correlated with that of sulfate and nitrate, respectively. Perhaps the time series in this study are so similar that it was not possible to separate each of them into individual factors, although this is surprising when OM is a large contributor to PM2.5 at all of the sites.

Section 3.2, lines 278-279: Please expand on exactly how the SOA is calculated here. Is it a percent of each of the sulfate and nitrate fractions based on the contribution of OM to each factor?

Section 3.2, line 292: What are the unidentified sources? Is it the residual from ME-2?

Section 3.3, lines 309-313: In other locations e.g. in Europe, secondary sulfate is typically a regional source so perhaps comment on whether it is typical in this area for sulfate to be a more locally influenced source. Also, regarding the correlations with meteorological conditions, mention that temperature also plays a role, especially in influencing ammonium nitrate concentrations.

Section 3.3, lines 324-326: This statement emphasizes the importance of running PMF as a first step for identifying a range of possible sources, both typical and atypical sources. It would be worthwhile indicating this in the manuscript.

Section 3.3, lines 330-332: Although in Tao's study, the ship emissions study may not be a pure primary source, it could still be representative of ship emissions in general even if it's more of a limitation of the PMF output. In contrast, in this study ME-2 was

used so sources are better separated yet the secondary sulfate and secondary nitrate factors comprise some organics. However, they are still likely secondary sources as the OA component is likely secondary also. Perhaps the naming of the secondary sulfate and secondary nitrate factors should be re-considered or clearly described in the text as being a predominantly secondary sulfate factor, for example.

Section 3.4, in general: The first part of this section, up to and possibly including Table 6, should be moved to earlier in the manuscript. Perhaps add it as a sub-section in the methods as a description of the different meteorological conditions and the links between wind direction, season, and monsoon. This would significantly help interpretations of the data earlier in the manuscript.

Section 3.4, lines 381-386: The average concentration of the ship emission source was similar between the two flows though. It is important to note this as the sources are referred to in terms of the average for the whole region during the different seasons/monsoons at other points in the manuscript.

Section 3.4, lines 417-419 and 443-444: The authors mention that the spatial distributions and source characteristics of secondary sulfate and secondary nitrate also reflected the corresponding characteristics of LV-OOA and SV-OOA, respectively. This is a circular point. The likely reason that each of the two sets of factors are not separated into individual factors is because the characteristics are similar between secondary sulfate and LV-OOA, for example. Temporally they will likely be the same and time-series are one of the main inputs for factor analysis. The sentences do not really make sense as the factor is a combination of the two, so of course they will show the same characteristics as there is only one output representing both sources. Please re-phrase and expand on this point. Similarly to an earlier comment on this, perhaps considering re-naming these factors would help reduce any confusion surrounding there not being separate LV-OOA and SV-OOA factors.

Section 3.4, lines 459-464: Have the authors considered the influence of residential

biomass burning? In other locations few/no coal-fired power stations, the biomass burning factors are typically associated with residential space heating and other residential activities. Are the coal-powered power stations here so dominant that residential biomass burning is negligible or is such an activity not typical in this region?

Figures and Tables:

Table 3, page 6: Please reduce the spacing of the factor names so that it's clear there are only four factors as it currently reads like there are six.

Figure 3a, page 8: Please add a line to the figure caption explaining the differences in size of the pie charts. Also, clarify that the number in brackets next to each of the abbreviated site names is the concentration; currently the figure caption only details the units.

Table 4, pages 8-9: There have been more recent studies in some of the locations detailed in the table e.g. the ClearfLo project in London spans several years, with 2012 being the main year of measurements, and there are several publications from this project alone. Perhaps other projects in these locations could be cited in the main text. Further, please explain and maybe add a sentence in the text as to why the particular studies are listed in the table for comparison e.g. the studies use similar methods and/or present similar results (in terms of the species measured) to allow for a better comparison with the current study.

Table 5, pages 12-13: Please rearrange this table so that the comparable studies in the same locations are next to each other i.e. group the studies in the table that were performed in Guangzhou etc. In addition, perhaps a small comment/note would be good on how the traffic source in the Huang et al study compares to the vehicle emissions source in this study (e.g. does one include tire/brake wear and the other doesn't).

Figure 9, page 17: Please add some more information such as a key or legend to the figure. For example, what are the triangles? A scale and N arrow would be useful

also. In the figure caption the 'shaded area' is noted as indicating the 'key emission area'. Firstly, perhaps a pattern could be used instead of red shading so as to prevent confusion that it represents the secondary sulfate source emission area. Secondly, please clarify if the 'key emission area' is for multiple different sources and reference the text in the manuscript where this is described further. Finally, the authors may wish to either move this figure to earlier in the manuscript or refer to it earlier in the text such as around line 380.

Supplement:

Table S2: Where did the 'PRD-annual' column come from? How were the numbers determined? It does not appear to be an average of the enrichment factors from the six sites listed in the rest of the table. Also, if the final column is based on the data collected in this study, then please add a note that the 'annual' is a estimation based on the four months of data collected during the study as opposed to 12 full months of measurements.

Figure S1: The image quality needs to be significantly improved. A key/legend, scale, lat/long details, and a N arrow should be added where possible and a couple of sentences explaining if the colors represent certain time periods, for example the purple/blues are for older dates and yellows are for newer dates (if a color-time scale is not available). Information in the caption needs to be added regarding the details of the trajectories themselves – are they 24-hour trajectories; were there any particular criteria entered for running them.

Figures S3 and S4: These figures need to be significantly improved to be clearer (currently they are fuzzy), include additional information such as the dates each of the six boxes represent and legends and scales. Some of these may be included in the small text boxes in the top left of each grid but these are currently not clear.

Figures S5 and S6: Similarly to the above, these figures need to be improved by sharpening the quality and clarity of the figures as well as including keys and scales where

possible.

Minor and technical corrections:

Abstract, line 15: Possible typographical error as the meaning of 'ever experience severe PM2.5 is not clear. Please rephrase.

Abstract, line 28: A space is needed between the end of 'burning' and the percentage '(11%)'.

Section 2.1, line 108: DRI has a new model analyzer so if possible please add a model number for the instrument used in this study assuming it is the older analyzer.

Section 2.2, line 122: A space is needed between 'F' and 'are'.

Section 2.3, line 172: Please define 'EV'. It is defined later in the manuscript but this is the first occurrence.

Section 3.1, line 202: Here it is stated that 'trace elements accounted for 6.2%' but figure 2 indicates that trace elements contribute only 1% and 'others' contribute the 6.2%. Which is the correct number?

Section 3.1, line 211: Possible typographical error as the meaning of 'the dominant northeastern wind the year' is not clear. Please address.

Section 3.1, line 225: Please explain what yellow label vehicles are.

Section 3.1, line 232: Please provide example references to the studies performed in each of the cities listed and/or refer to table 4 where there are references.

Section 3.3, line 323: A space is needed between 'years' and '(People's Government'.

Section 3.4, lines 355-356: Please briefly comment on the other types of flows e.g. easterly flow?

Section 3.4, lines 390-393: Is there something that can be used as further evidence or to reference the road construction noted here or is it based on local knowledge?

Section 3.4, lines 426-427: Show the coal-fired power plants on the map in figure 9.

Section 3.2, line 444: possible typographical error: this is meant to be SV-OOA instead of LV-OOA.

Section 4, line 519: Was this meant to read 'in recent decades' i.e. plural decades?
* * *

---

## Referee Comment (RC2) · Anonymous Referee #2 · 30 Apr 2018

General comments:

This study apportioned the sources of fine particles in the Pearl River Delta (PRD) region of China using both PMF version 5 and ME-2 methods. The authors found that ME-2 model could produce better results than the PMF model. Ten sources of PM2.5 were found in the PRD region including secondary sulfate (21%), vehicle emissions (14%), industrial emissions (13%), secondary nitrate (11%), biomass burning (11%), SOA (7%), coal combustion (6%), fugitive dust (5%), ship emission (3%), and aged sea salt (2%). Furthermore, authors identified the source contribution from both local and regional emissions.

[Figure]

In general, the scientific content in this manuscript is good for publication. However, I have some comments that I hope it could help author improve their manuscripts.

Major comments:

1, Line 109: The authors assumed OM/OC is 1.8. This ratio seems too high for me. According to He et al. (2011), the OM/OC is 1.6 for the urban areas. Could the author explain for this ratio? In addition, why do you use the OM, not OC as the input variable in the model? I think OM/OC ratios should vary following the sampling days. Therefore, if you input the OM instead of OC in the model, it will cause more uncertainties. How did the authors calculate the uncertainty for the OM?

2, PMF model vs ME-2 This study compared the PMF and ME-2, but I cannot find the information which shows how the authors conducted the PMF in details. I suggested that the author should write more about PMF version 5.0, what is difference between PMF v5.0 and ME-2. For example, in PMF v.5, they also have constrained factor functions, did the authors use this function to constrain the factor? In addition, the authors should write more how they select the number of the factors and optimize the PMF results. I would be grateful if the authors show correlations between the PMF and ME-2 results.

Line 164: Qtrue/Qexp =2.5. Could the authors explain why they use the Qtrue/Qexp ratio of 2.5 to optimize the solution? I think the ratios depend on the number of factors and the uncertainties. Did the author add the extra uncertainty in the PMF model?

Other minor comments:

1, Line 172: Please define "EV"

2, Line 205-206: I think the much lower concentration of PM2.5 at DP because this sampling site near the sea therefore the air pollutants are more diluted. I am not really clear why low PM2.5 concentration at DP indicate the large contributions of pollution transported from outside region? Could the author explain for this?

3, Line 227-230: The authors compared the PM2.5 between the cities. This comparison is not meaningful to me because the authors compared the levels at different time periods. For example the PM2.5 levels at Beijing and Tianjin were measured in 2012-2013, while the PM2.5 concentration measured in this study was in 2015. Please note that after 2012, the PM2.5 trends at Beijing and Tianjin also showed a huge decrease under the "Control Action Plan". I suggest the author should update the PM2.5 level in the Table 4.

4, Line 252-256: Could the author explain "high OM concentration was considered to present the LV-OOA" and "high OM concentration was considered to represent SV-OOA"? Could you please discuss more about that: why the (NH4)2SO4 associated with LV-OOA and NH4NO3 and SV-OOA shared same source?

5, Figure 8: Regarding the aged sea-salt factor, the contribution of this factor at QA and HS sites from the northerly flow was higher than those from the southerly flow. Could you explain for that?

6, Line 527: A typo-mistake "theMe-2".

---

## Author Comment (AC1) · 25 Jun 2018

General comments:

Huang et al. investigate the sources of PM2.5 in the Pearl River Delta (PRD) region of China, determining whether the sources are local or regional and how they vary under different meteorological conditions based on six sites representing urban, suburban, and background locations. The authors present detailed chemical composition results from data collected at each of the sites for approximately one month during each the main four seasons to represent the variability during a full calendar year. Both PMF and ME-2 were applied to the data to identify potential sources of PM2.5 in the area, which

were subsequently correlated with meteorological conditions, such as monsoons, to further identify the importance of each of the sources including local versus regional nature and temporal significance. The authors compared the findings from this work with previous studies in the same areas as well as putting the results into a global context. Ultimately, the authors were able to identify key emission sources and locations that should be targeted in future pollution control measures.

Although the scientific quality of the work is good, the authors do not obviously highlight the uniqueness of this study. The data presented are new thus add to the scientific knowledge and understanding of the PRD and the methods used, particularly ME-2, appear to be novel in that they are applied to a unique dataset. If this is the case, the authors should include a sentence or two in the appropriate places within the manuscript (e.g. abstract). The scientific methods and assumptions are valid and the results are generally sufficient to support the interpretations and conclusions although some additional evidence or explanation is needed (see specific comments). Numerous pie charts are presented in the manuscript; the authors should consider moving some of these to the supplementary material to reduce the length of the manuscript or use a different style plot to distinguish between the different types of results being presented. The figures currently in the supplementary material need to be greatly improved in terms of clarity of the images as well as the addition of legends where possible. The manuscript generally flowed well but it could do with some slight reordering, especially the section describing the meteorological conditions, to make the manuscript flow even better.

Despite there being some major points that need to be revised, the overall quality of the work presented and manuscript itself are good; an interesting and enjoyable read. Once the revisions above and the comments below are addressed, I recommend this manuscript be published in ACP.

Reply:

Many thanks for the kind and helpful comments of this reviewer. The general comments above have all been solved in the revised manuscript. Please refer to the reply to the corresponding specific comments below.

Specific comments:

Abstract, line 21: It is stated that the 'regional annual average PM2.5 concentration was determined'. This is misleading as there were only ~4 months of measurements spanning the year, with samples taken every other day. Although those four months may be representative of the main pollution conditions, it should be mentioned or clarified that a full year of data was not obtained to determine/estimate this annual average.

Reply:

The sentence has been supplemented with "...based on the 4-month sampling".

Introduction, lines 50-51: Why are these noteworthy provinces? Please consider adding a few words as to why these are being highlighted.

Reply:

Sorry, it is a typographical error. They are nine cities, not provinces. The PRD region consists of these 9 cities.

Introduction, line 62: It is stated that the previous studies in the PRD provinces 'lacked the extensive representation of the PRD'. Please qualify this statement e.g. is it because only single locations were studied and that is the uniqueness of this study as several locations are studied at the same time.

Reply:

Corrected to: "However, the above source apportionment studies only focused on part of PM2.5 (e.g., organic matter) or single city in PRD (e.g., Shenzhen and Dongguan), lacking the extensive representation of the PRD region in terms of simultaneous sampling in multiple cities."

Introduction, lines 65-67: Despite some of PMF's limitations, it is the first step for the application of ME-2 to a dataset. Further, as PMF does not require a prior information, new sources could be identified as a result both in terms of newly identified as a source in a given location or a newly identified emission source overall. Please add a sentence or two to acknowledge that PMF is usually the first step in factor analysis using ME-2, especially as the a priori information used for running ME-2 typically uses the factor profiles identified from PMF and/or identifies a number of factors that should be considered when running ME-2.

Reply:

Suggestion taken. The following sentence has been added: "The key challenges in running ME-2 are the construction of the appropriate constraint source profiles and the determination of factor numbers, and PMF could serve as the first step when using ME-2 for the determination of the priori information needed."

Introduction, lines 69-74: As mentioned, organic aerosols have been successfully apportioned using ME-2 via SoFi. As this study uses both organic and inorganic species it would be good to point this out as being novel. If it is the first study of its kind to apply the model to this dataset (in terms of the species and/or measurement period and location) then this should be highlighted in the manuscript in the appropriate places such as the abstract and later in the introduction. If this is not a unique case then something along the lines above should be mentioned in any case along with a citation of similar cases for comparison.

Reply:

In abstract, revised to: "A novel multilinear engine (ME-2) model was firstly applied to a comprehensive PM2.5 chemical dataset to perform source apportionment with predetermined constraints..."

In introduction, revised to: "For the first time, the novel ME-2 model via the SoFi was

applied to a comprehensive chemical dataset (including EC, OM, inorganic ions and metal elements) to identify the sources of bulk PM2.5 in the regional scale of PRD...".

Section 2.1, lines 93-94: The current way in which the sampling periods are described are misleading as 'January-February' could be interpreted as being two full months whereas in fact it is a period of one month spanning two months. Add a few words clarifying that each sampling period for the seasons is one month and refer to table 2, where the exact sampling dates are noted.

Reply:

Corrected to: "Samples were collected every other day during a one-month long period for each season in 2015, and Table 2 contains the detailed sampling information to refer to."

Section 2.1, lines 100-101: 'two different types of samplers sampled' – clarify that it is the two samplers that were used in this study that were compared. The results of the inter-instrument comparison 'yielded a relative deviation of less than 5% for PM2.5 mass concentrations'. How many samples were obtained for this comparison? How was the 5% calculated/determined? Please consider adding something to the manuscript on this.

Reply:

Corrected to: "Prior to the sampling campaigns, the six samplers used sampled in parallel for three times, and each time lasted for 12 h. The standard deviation of the PM2.5 mass concentrations obtained by the six samplers in each parallel sampling was within 5%."

Section 2.1, in general: There is no mention of the exact number of samples that were obtained and whether there were any issues with any of them. Are all âˇ15 samples from each season valid and run as intended? What QA/QC was performed on the samples (standard laboratory QC and overall QA)?

[Figure]

Reply:

The following information has been added. "After each sampling, the Teflon filters were put into Poly tetra fluoroethylene (PTFE) boxes and the Quartz filters were put into PTFE boxes with 500 °C burned aluminum foil inside. The sample boxes were then sealed by Parafilm, stored in an ice-packed cooler during transportation, and stored under freezing temperatures before analysis. A total of 362 valid samples (15-16 samples at each site for each season) were collected in this study. In addition, to track the possible contamination caused by the sampling treatment, a field blank sample was collected at each site for each season. The PM2.5 mass can be obtained based on the difference in the weight of the Teflon filter before and after sampling in a cleanroom at conditions of 20°Cand 50% relative humidity, according to the QA/QC procedures of the National Environmental Protection Standard (NEPS, MEE, 2013a). The Teflon filters were analyzed for their major ion contents (SOâĆĎ$^2$ËĽ, NOâĆČËĽ, NHâĆĎâĄž and ClËĽ) via an ion chromatography system (ICS-2500, Dionex; Sunnyvale, California, USA), following the guidelines of NEPS (MEE, 2016a,b). The metal element contents (23 species) were analyzed via an inductively coupled plasma mass spectrometer (ICP-MS, auroraM90; Bruker, Germany), also following the guidelines of NEPS (MEE, 2013b). The Quartz filters were analyzed for organic carbon (OC) and elemental carbon (EC) contents using an OC/EC analyzer (2001A, Desert Research Institute, Reno, Nevada, USA), following the IMPROVE protocol (Chow et al., 1993)."

Section 2.2, lines 146-147: Please provide examples of the 'sources of uncertainty that contributed little to the total uncertainty'.

Reply: Examples are now given as below: "...such as replacing filters, sample transport and sample storage under the strict QA/QC."

Section 2.2, lines 159-160: Please expand on why a factor of 2 was applied to the estimated uncertainties. Specifically, please explain why a factor of 2 was chosen. If this is this a typical factor to apply, please provide a reference.

Reply: The following information is now added:

"The uncertainties of SO₄²⁻, NH₄⁻ and all metal elements, which have scaled residuals larger than ±3 due to the small analytical uncertainties in Table S3, need to be increased to reduce their weights in the solution (Norris et al., 2014). In addition, the uncertainties of EC caused by pyrolyzed carbon (PC), the uncertainties of OM, NO₃⁻ and Cl⁻ due to semi-volatility under high ambient temperatures should also be taken into account (Cao et al., 2017). In this study, more reasonable source profiles can be obtained when further increasing the estimated uncertainties (u $\check{I}$_c) of all species by a factor of 2."

Section 2.3, line 168: It is not typical to refer to later sections in a manuscript. Perhaps consider summarizing what is in the later sections here or rephrase this sentence so that Section 3.2 can be referenced but the reader does not have to read that section at this point.

Reply: Rephrased to:

"For the nine-factor solution of secondary sulfate-rich, secondary nitrate-rich, aged sea salt, fugitive dust, biomass burning, vehicle emissions, coal burning, industrial emissions and ship emissions, the source judgement based on tracers for each factor was identical to that of the ME-2 results detailed in Section 3.2."

Section 3.1., lines 211-215: Figure 3b does not show that the seasonal variations in the major PM2.5 components were correlated with monsoon characteristics. Please expand on this, clarify, and/or provide additional evidence for this statement. Similarly please expand on/clarify how figure S1 shows that the northern monsoon prevails in winter and the southern monsoon prevails in summer.

Reply:

Figure S1 was replaced with clustered back trajectories, showing that the northern monsoon (94%) prevailed in winter and the southern monsoon (78%) prevailed in summer. The sentences are rephrased to: "The back trajectories of the air masses (Fig. S1) show that the northern monsoon prevails in winter and the southern monsoon prevails in summer in the PRD. Under the winter monsoon, the air masses mostly came from the inland and carried higher concentrations of air pollutants. However, under the summer monsoon, the air masses largely originated from the South China Sea and were clean. In addition, the frequent rainfall and higher planetary boundary layer (PBL) in summer in the PRD also favored the dispersion and removal of air pollutants (Huang et al., 2014b). Fig. 3b shows that the normalized seasonal variations of the major components in PM2.5 in the PRD were evidently higher in winter and lower in summer, well consistent with the seasonal variations of monsoon and other meteorological factors as mentioned above."

Section 3.2, lines 251-257: Please comment on why it might not have been possibly to separate the secondary sulfate and LV-OOA as two separate factors as well as SVOOA and secondary nitrate. Having a mixed factor is something the authors note as being a downfall of the PMF results so it needs to be acknowledged that even with SoFi there is a mixed factor. To confirm LV-OOA/OOA-1 and SV-OOA/OOA-2 factors in other studies, the time series is correlated with that of sulfate and nitrate, respectively. Perhaps the time series in this study are so similar that it was not possible to separate each of them into individual factors, although this is surprising when OM is a large contributor to PM2.5 at all of the sites.

Reply:

Comments added as below: "In this study, secondary organic aerosol (SOA) did not appear as a single factor, even if we run the ME-2 with ten or more factors. SOA can usually be described by low-volatile oxygenated organic aerosol (LV-OOA) and semi-volatile oxygenated organic aerosol (SV-OOA), based on the volatility and oxidation state of organics (Jimenez et al., 2009). In previous studies (e.g., He et al., 2011; Lanz et al., 2007; Ulbrich et al., 2009), the time series of LV-OOA and SV-OOA were highly correlated with those of sulfate and nitrate, respectively, implying that LV-OOA and sulfate (or SV-OOA and nitrate) cannot be separated easily in cluster analysis, especially when there is no effective tracer of SOA. In this study, the high OM concentration in the secondary sulfate-rich factor was considered to represent LV-OOA, while the high OM concentration in the secondary nitrate-rich factor was considered to represent SV-OOA (Yuan et al., 2006b; He et al., 2011). Therefore, it should be acknowledged that mixed secondary factors cannot be solved even using ME-2."

Section 3.2, lines 278-279: Please expand on exactly how the SOA is calculated here. Is it a percent of each of the sulfate and nitrate fractions based on the contribution of OM to each factor?

Reply: Information added as below:

"In this study, however, an SOA factor can be reasonably extracted from the secondary sulfate-rich and secondary nitrate-rich factors and regarded as the sum of the OM concentrations in these two factors, i.e., LV-OOA+SV-OOA, leaving the remaining mass as independent secondary sulfate and secondary nitrate."

Section 3.2, line 292: What are the unidentified sources? Is it the residual from ME-2?

Reply:

The unidentified source is the difference between the total PM2.5 mass weighted and the total identified sources by ME-2, and includes both the residual from ME-2 and the unmeasured species. This information has been added into the sentence.

Section 3.3, lines 309-313: In other locations e.g. in Europe, secondary sulfate is typically a regional source so perhaps comment on whether it is typical in this area for sulfate to be a more locally influenced source. Also, regarding the correlations with meteorological conditions, mention that temperature also plays a role, especially in influencing ammonium nitrate concentrations.

Reply:

[Figure]

Sulfate is also a regional species in PRD, although it has big seasonal variation. We did not intend to regard sulfate as a local pollutant. We have rephrased the sentences to make the point not misleading as below, with the role of temperature mentioned: "The contributions of most sources were higher in winter and lower in summer, e.g., secondary sulfate, secondary nitrate, fugitive dust, biomass burning, vehicle emissions, coal burning, industrial emissions and SOA; these sources were greatly influenced by the seasonal variations of monsoon, rainfall and PBL, as discussed in Section 3.1. For example, although secondary sulfate was proven to be a typical regional pollutant in the PRD (Huang et al., 2014b; Zou et al., 2017), the more polluted continental air mass in the winter monsoon made its concentrations in winter much higher than in summer. The semi-volatile secondary ammonium nitrate was also significantly affected by seasonal ambient temperatures. In contrast, the contributions of aged sea salt and ship emissions displayed little seasonal variations, consistent with that the emissions were from local surrounding sea areas."

Section 3.3, lines 324-326: This statement emphasizes the importance of running PMF as a first step for identifying a range of possible sources, both typical and atypical sources. It would be worthwhile indicating this in the manuscript.

Reply:

Agree to this point. We have pointed out this in the introduction part as "The key challenges in running ME-2 are the construction of the appropriate constraint source profiles and the determination of factor numbers, and PMF could serve as the first step when using ME-2 for the determination of the priori information needed."

Section 3.3, lines 330-332: Although in Tao's study, the ship emissions study may not be a pure primary source, it could still be representative of ship emissions in general even if it's more of a limitation of the PMF output. In contrast, in this study ME-2 was used so sources are better separated yet the secondary sulfate and secondary nitrate factors comprise some organics. However, they are still likely secondary sources as

the OA component is likely secondary also. Perhaps the naming of the secondary sulfate and secondary nitrate factors should be re-considered or clearly described in the text as being a predominantly secondary sulfate factor, for example.

Reply:

Suggestion taken. The factor containing secondary sulfate and LV-OOA has been re-named as "secondary sulfate-rich", and the factor containing secondary nitrate and SV-OOA has been renamed as "secondary nitrate-rich".

Section 3.4, in general: The first part of this section, up to and possibly including Table 6, should be moved to earlier in the manuscript. Perhaps add it as a sub-section in the methods as a description of the different meteorological conditions and the links between wind direction, season, and monsoon. This would significantly help interpretations of the data earlier in the manuscript.

Reply:

Suggestion taken. Moved to section 2.2.

Section 3.4, lines 381-386: The average concentration of the ship emission source was similar between the two flows though. It is important to note this as the sources are referred to in terms of the average for the whole region during the different seasons/monsoons at other points in the manuscript.

Reply:

Suggestion taken. It has been noted in the manuscript as "the average contributions of aged sea salt and ship emissions for the whole region displayed little seasonal variations..."

Section 3.4, lines 417-419 and 443-444: The authors mention that the spatial distributions and source characteristics of secondary sulfate and secondary nitrate also reflected the corresponding characteristics of LV-OOA and SV-OOA, respectively. This is

a circular point. The likely reason that each of the two sets of factors are not separated into individual factors is because the characteristics are similar between secondary sulfate and LV-OOA, for example. Temporally they will likely be the same and time-series are one of the main inputs for factor analysis. The sentences do not really make sense as the factor is a combination of the two, so of course they will show the same characteristics as there is only one output representing both sources. Please re-phrase and expand on this point. Similarly to an earlier comment on this, perhaps considering re-naming these factors would help reduce any confusion surrounding there not being separate LV-OOA and SV-OOA factors.

Reply:

We have renamed the mixed factors as "secondary sulfate-rich" and "secondary nitrate-rich", and rephrased the sentences as below: "Since both secondary sulfate and LV-OOA belong to a mixed factor with fixed proportions, the spatial distribution of secondary sulfate also reflects the corresponding characteristics of LV-OOA.". "Since both secondary sulfate and LV-OOA belong to a mixed factor with fixed proportions, the spatial distribution of secondary sulfate also reflects the corresponding characteristics of LV-OOA."

Section 3.4, lines 459-464: Have the authors considered the influence of residential biomass burning? In other locations few/no coal-fired power stations, the biomass burning factors are typically associated with residential space heating and other residential activities. Are the coal-powered power stations here so dominant that residential biomass burning is negligible or is such an activity not typical in this region?

Reply:

The original expression "the frequent open-burning of crop residues" is not comprehensive. We have corrected it to "the popular events of open burning and residential burning of biomass wastes."

Figures and Tables:

Table 3, page 6: Please reduce the spacing of the factor names so that it's clear there are only four factors as it currently reads like there are six.

Reply: Corrected.

Figure 3a, page 8: Please add a line to the figure caption explaining the differences in size of the pie charts. Also, clarify that the number in brackets next to each of the abbreviated site names is the concentration; currently the figure caption only details the units.

Reply: Suggestion taken.

Table 4, pages 8-9: There have been more recent studies in some of the locations detailed in the table e.g. the ClearfLo project in London spans several years, with 2012 being the main year of measurements, and there are several publications from this project alone. Perhaps other projects in these locations could be cited in the main text. Further, please explain and maybe add a sentence in the text as to why the particular studies are listed in the table for comparison e.g. the studies use similar methods and/or present similar results (in terms of the species measured) to allow for a better comparison with the current study.

Reply:

After careful examining the literature, we found that the publications on PM2.5 from the ClearfLo project in London only focused on trace elements based on filter samples. We cannot obtain results of similar species like OC, EC and SIA in this study. We have updated this table with more recent studies in Beijing, Shanghai and Chengdu in China and Chuncheon in Korea.

Comment for the selection of the studies in the table has been added as below: "Table 5 summarizes some previous studies that used similar filter-sampling and analytical methods to allow for a better comparison with this study."

[Figure]

Table 5, pages 12-13: Please rearrange this table so that the comparable studies in the same locations are next to each other i.e. group the studies in the table that were performed in Guangzhou etc. In addition, perhaps a small comment/note would be good on how the traffic source in the Huang et al study compares to the vehicle emissions source in this study (e.g. does one include tire/brake wear and the other doesn't).

Reply:

Table rearranged. In fact, the direct comparison of traffic emissions between this study and Huang et al. (2014a) may not be significant due to different species input into different models. Especially, the traffic source profile in Huang et al. (2014a) contains a large fraction of unidentified mass.

Figure 9, page 17: Please add some more information such as a key or legend to the figure. For example, what are the triangles? A scale and N arrow would be useful also. In the figure caption the 'shaded area' is noted as indicating the 'key emission area'. Firstly, perhaps a pattern could be used instead of red shading so as to prevent confusion that it represents the secondary sulfate source emission area. Secondly, please clarify if the 'key emission area' is for multiple different sources and reference the text in the manuscript where this is described further. Finally, the authors may wish to either move this figure to earlier in the manuscript or refer to it earlier in the text such as around line 380.

Reply: All Suggestions taken.

Supplement:

Table S2: Where did the 'PRD-annual' column come from? How were the numbers determined? It does not appear to be an average of the enrichment factors from the six sites listed in the rest of the table. Also, if the final column is based on the data collected in this study, then please add a note that the 'annual' is a estimation based on the four months of data collected during the study as opposed to 12 full months of

[Figure]

measurements.

Reply:

The "PRD-annual" is based on the average of the spring, summer, autumn and winter samples of the six sites. "PRD-annual" has been replaced with "Average of four months at six sites".

Figure S1: The image quality needs to be significantly improved. A key/legend, scale, lat/long details, and a N arrow should be added where possible and a couple of sentences explaining if the colors represent certain time periods, for example the purple/blues are for older dates and yellows are for newer dates (if a color-time scale is not available). Information in the caption needs to be added regarding the details of the trajectories themselves – are they 24-hour trajectories; were there any particular criteria entered for running them.

Reply:

Figure S1 has been replaced by clustered back trajectories. All suggestions taken for the updated figure.

Figures S3 and S4: These figures need to be significantly improved to be clearer (currently they are fuzzy), include additional information such as the dates each of the six boxes represent and legends and scales. Some of these may be included in the small text boxes in the top left of each grid but these are currently not clear.

Reply: Corrected. The figure scales cannot be obtained at the original website (http://www.hko.gov.hk/wxinfo/currwx/wxchtc.htm).

Figures S5 and S6: Similarly to the above, these figures need to be improved by sharpening the quality and clarity of the figures as well as including keys and scales where possible.

Reply: Corrected.

Minor and technical corrections:

Abstract, line 15: Possible typographical error as the meaning of 'ever experience severe PM2.5 is not clear. Please rephrase.

Reply: Corrected to "and had severe PM2.5 pollution at the beginning of this century."

Abstract, line 28: A space is needed between the end of 'burning' and the percentage '(11%)'.

Reply: Corrected.

Section 2.1, line 108: DRI has a new model analyzer so if possible please add a model number for the instrument used in this study assuming it is the older analyzer.

Reply: The model number has been added as "2001A, Desert Research Institute, Reno, Nevada, USA".

Section 2.2, line 122: A space is needed between 'F' and 'are'.

Reply: Corrected.

Section 2.3, line 172: Please define 'EV'. It is defined later in the manuscript but this is the first occurrence.

Reply: Corrected.

Section 3.1, line 202: Here it is stated that 'trace elements accounted for 6.2%' but figure 2 indicates that trace elements contribute only 1% and 'others' contribute the 6.2%. Which is the correct number?

Reply: Corrected. Trace elements accounted for 1.0%.

Section 3.1, line 211: Possible typographical error as the meaning of 'the dominant northeastern wind the year' is not clear. Please address.

Reply: Corrected to "under the northeastern wind, which is the most frequent wind in

the PRD".

Section 3.1, line 225: Please explain what yellow label vehicles are.

Reply: Corrected to "older and more polluting vehicles".

Section 3.1, line 232: Please provide example references to the studies performed in each of the cities listed and/or refer to table 4 where there are references.

Reply: Suggestion taken. Revised to "Paris (Bressi et al., 2013), London (Rodríguez et al., 2007), and Los Angeles (Hasheminassab et al., 2014), while they were similar to those of Santiago (Villalobos et al., 2015) and Chuncheon (Cho et al., 2016)".

Section 3.3, line 323: A space is needed between 'years' and '(People's Government'.

Reply: Corrected.

Section 3.4, lines 355-356: Please briefly comment on the other types of flows e.g. easterly flow?

Reply: Suggestion taken. Revised to "Southerly flow and northerly flow appeared with the highest frequency in the PRD (i.e., above 80%), followed by cyclone (10%), easterly (2%) and trough (2%)."

Section 3.4, lines 390-393: Is there something that can be used as further evidence or to reference the road construction noted here or is it based on local knowledge?

Reply: An official evidence has been added as "…while the high value at QA under northerly flow maybe related to the reconstruction project of the adjacent Nansha Port (Guangzhou Municipal People's Government, 2015)."

Section 3.4, lines 426-427: Show the coal-fired power plants on the map in figure 9.

Reply: The PRD has many coal-fired power plants. We tried but failed in getting enough information of the power plants.

Section 3.2, line 444: possible typographical error: this is meant to be SV-OOA instead

of LV-OOA.

Reply: Corrected.

Section 4, line 519: Was this meant to read 'in recent decades' i.e. plural decades?

Reply: Corrected to "in the past ten years".

---

## Author Comment (AC2) · 25 Jun 2018

General comments: This study apportioned the sources of fine particles in the Pearl River Delta (PRD) region of China using both PMF version 5 and ME-2 methods. The authors found that ME-2 model could produce better results than the PMF model. Ten sources of PM2.5 were found in the PRD region including secondary sulfate (21%), vehicle emissions (14%), industrial emissions (13%), secondary nitrate (11%), biomass burning (11%), SOA (7%), coal combustion (6%), fugitive dust (5%), ship emission (3%), and aged sea salt (2%). Furthermore, authors identified the source contribution from both local and regional emissions.

[Figure]

In general, the scientific content in this manuscript is good for publication. However, I have some comments that I hope it could help author improve their manuscripts.

Major comments:

1, Line 109: The authors assumed OM/OC is 1.8. This ratio seems too high for me. According to He et al. (2011), the OM/OC is 1.6 for the urban areas. Could the author explain for this ratio? In addition, why do you use the OM, not OC as the input variable in the model? I think OM/OC ratios should vary following the sampling days. Therefore, if you input the OM instead of OC in the model, it will cause more uncertainties. How did the authors calculate the uncertainty for the OM?

Reply:

We agree that the OM/OC should vary to some extent from sample to sample, although this ratio is difficult to measure and usually fixed at a constant. However, an advantage of fixing the OM/OC at a constant is that additional uncertainty can be avoided in the transformation from OC to OM, since the columns of G (factor time series) are normalized in the model calculation process (Paatero et al., 1994). Thus, it is the same using OM or OC in the model. In previous aerosol mass spectrometry measurement for PM1, the OM/OC ratio was measured to be 1.6 for urban atmosphere (He et al., 2011) and 1.8 for rural atmosphere (Huang et al., 2011), we adopted 1.8 for the six sites (including urban, suburban, and background atmospheres) because it is assumed that the difference between PM1 and PM2.5 may contain more aged regional aerosol with higher OM/OC, which has been explained in the revised text.

2, PMF model vs ME-2 This study compared the PMF and ME-2, but I cannot find the information which shows how the authors conducted the PMF in details. I suggested that the author should write more about PMF version 5.0, what is difference between PMF v5.0 and ME-2. For example, in PMF v.5, they also have constrained factor functions, did the authors use this function to constrain the factor? In addition, the authors should write more how they select the number of the factors and optimize the

[Figure]

PMF results. I would be grateful if the authors show correlations between the PMF and ME-2 results.

Reply:

More details of the PMF running have been provided as below: "After examining a range of factor numbers from 3 to12, the 9-factor solution output by the PMF base run (Qtrue/Qexp=2.5) was found to be the optimal solution, with the scaled residuals approximately symmetrically distributed between –3 and +3 (Fig. S6) and the most interpretable factor profiles (Fig. S7). The model-input total mass of the 18 species and the model-reconstructed total mass of all the factors showed a high correlation (R2=0.97, slope=1.01) (Fig. S8). The factor of biomass burning was not extracted in the eight-factor solution, while the factor of fugitive dust was separated into two non-meaningful factors when more factors were set to run PMF."

More descriptions about the difference between PMF and ME-2 are added as below: "SoFi is a user-friendly interface developed by PSI for initiating and controlling ME-2 (Canonaco et al., 2013), and it can conveniently constrain multiple factor profiles. Although USEPA PMF v5.0 can also use some priori information (such as ratio of elements in factor) to control the rotation after the base run, it is not able to use multiple constrained factor profiles to control the rotation (Norris et al., 2014). Therefore, SoFi is a more convenient and powerful tool to establish various constrained factors for source apportionment modeling."

A comment is added for the comparison between PMF and ME-2 results in Section 3.2ïijŽ "Although these nine factors of the ME-2 modeling generally showed high correlations (R2=0.81–0.97) with the corresponding factors of the PMF modeling in terms of time series, it is easy to see that the ME-2 modeling provided a better..."

Line 164: Qtrue/Qexp =2.5. Could the authors explain why they use the Qtrue/Qexp ratio of 2.5 to optimize the solution? I think the ratios depend on the number of factors and the uncertainties. Did the author add the extra uncertainty in the PMF model?

Reply:

Yes, the Qtrue/Qexp ratio depends on the number of factors and the uncertainties. Ideally, if the model entirely captured the variability of the measured data and all uncertainties were properly defined, a Qtrue/Qexp value of 1 would be expected. We did not intend to say the Qtrue/Qexp ratio of 2.5 is the best value, but intend to monitor this value and compare it to that of the ME-2 solution. In result, the ratio of the ME-2 solution (1.2) is closer to 1.0, indicating that the species residuals had decreased and the ME-2 solution should be more reasonable. Extra uncertainties in this study were added as below: "The uncertainties of $SO_4^{2-}$, $NH_4^+$ and all metal elements, which have scaled residuals larger than $\pm 3$ due to the small analytical uncertainties in Table S3, need to be increased to reduce their weights in the solution (Norris et al., 2014)." The above points have been clarified in the revised manuscript.

Other minor comments:

1, Line 172: Please define "EV"

Reply: Suggestion taken.

2, Line 205-206: I think the much lower concentration of PM2.5 at DP because this sampling site near the sea therefore the air pollutants are more diluted. I am not really clear why low PM2.5 concentration at DP indicate the large contributions of pollution transported from outside region? Could the author explain for this?

Reply:

To make the point clearer, we have rephrased the text as below: "The DP background site had little local emission and was hardly influenced by the emissions from the PRD under both southerly flow and northerly flow. Thus, its air pollution reflects the large-scale regional air pollution. The average PM2.5 concentration at DP was as high as 28 $\mu$g/m$^3$, indicating that the PRD had a large amount of air pollution transported from outside this region."

3, Line 227-230: The authors compared the PM2.5 between the cities. This comparison is not meaningful to me because the authors compared the levels at different time periods. For example the PM2.5 levels at Beijing and Tianjin were measured in 2012-2013, while the PM2.5 concentration measured in this study was in 2015. Please note that after 2012, the PM2.5 trends at Beijing and Tianjin also showed a huge decrease under the "Control Action Plan". I suggest the author should update the PM2.5 level in the Table 4.

Reply:

Updated with more recent data available in the literature.

4, Line 252-256: Could the author explain "high OM concentration was considered to present the LV-OOA" and "high OM concentration was considered to represent SVOOA"? Could you please discuss more about that: why the (NH4)2SO4 associated with LV-OOA and NH4NO3 and SV-OOA shared same source?

Reply:

The following discussion has been added into the revised manuscript. "In this study, secondary organic aerosol (SOA) did not appear as a single factor, even if we run the ME-2 with ten or more factors. SOA can usually be described by low-volatile oxygenated organic aerosol (LV-OOA) and semi-volatile oxygenated organic aerosol (SV-OOA), based on the volatility and oxidation state of organics (Jimenez et al., 2009). In previous studies (e.g., He et al., 2011; Lanz et al., 2007; Ulbrich et al., 2009), the time series of LV-OOA and SV-OOA were highly correlated with those of sulfate and nitrate, respectively, implying that LV-OOA and sulfate (or SV-OOA and nitrate) cannot be separated easily in cluster analysis, especially when there is no effective tracer of SOA. In this study, the high OM concentration in the secondary sulfate-rich factor was considered to represent LV-OOA, while the high OM concentration in the secondary nitrate-rich factor was considered to represent SV-OOA (Yuan et al., 2006b; He et al., 2011)."

5, Figure 8: Regarding the aged sea-salt factor, the contribution of this factor at QA and HS sites from the northerly flow was higher than those from the southerly flow. Could you explain for that?

Reply:

The following discussion has been added. "The spatial distribution of aged sea salt among the different sites was a complex result of the site locations relative to the sea and meteorological conditions, e.g., wind and tide. A relatively high level of aged sea salt was observed at the Qi-Ao Island (QA), especially in the northerly flow, which can be attributed to that the QA site was surrounded by the sea and had lower wind speeds in the northerly flow (in Table 3)."

6, Line 527: A typo-mistake "theMe-2".

Reply: Corrected.

---

## Author Response (AR2)

General comments:

The authors have responded to the comments from both reviewers and although the revisions were not major from either reviewer, the authors have improved the manuscript in multiple places. A couple of small revisions are recommended before final publication in ACP.

Specific comments:

Page 7, line 223: Please explain either here or in the supplement what explained variations mean/represent so that it is clear whether a low or high percentage is preferable for the chosen factor.

Reply:

Suggestion taken. Corrected to 'had obvious percentage explained variation (EV) values, i.e., the percent of a species apportioned to the factor'.

Page 7, lines 221-227: This section is not clear in terms of how the values presented as EVs compare to the figures. In the figures (e.g. Fig. S7), concentration and percent of species are plotted but EV is not.

Reply:

As in the reply to the question above, we have defined that EV is just the percent of species in Figure S7, and we have also clarified this definition in the caption of Figure S7.

Page 11, lines 354-357: The authors should clarify how the SOA factor is extracted. Perhaps clarify by adding text to the supplement or a brief sentence or two in the main text.

Reply:

Rephrased to: 'However, the contribution time-series of LV-OOA (or SV-OOA) can be extracted based on the contribution time-series of the secondary sulfate-rich factor (or the secondary nitrate-rich factor) and the mass percentage of OM in this factor, leaving the remaining mass as the "pure" secondary sulfate (or secondary nitrate). Therefore, a new SOA factor can be reasonably estimated by LV-OOA+SV-OOA.'

Page 20, lines 581-584: The authors may wish to clarify in the conclusions that the four months of sampling were conducted to represent each of the four seasons during the year thus representing the annual PM2.5 pollution.

Reply:

Corrected to: 'to conduct four months (one for each season) of sampling…'

Minor and technical corrections:

Page 4, lines 104-105: Remove 'for' so the sentence reads 'the six samplers used sampled in parallel three times'.

Reply:
Corrected.

Page 4, lines 106-108: Please rephrase this sentence as it is currently not clear what is meant. Were Teflon filters put into boxes made of the same material? Perhaps the sentence can be simplified or even removed unless this step is critical to the sampling handling process.

Reply:
Suggestion taken. This sentence is not important and is now removed.

Page 4, line 115: The authors may want to consider writing QA/QC out in full here and elsewhere in the manuscript/supplement to avoid confusion with the 'QA' used for the Qi-Ao island site.

Reply:
Suggestion taken. The 'QA/QC' has been replaced with 'Quality Assurance and Quality Control'.

Page 7, line 210: A space is needed between 'to' and '12' to read 'factor numbers from 3 to 12...'.

Reply:
Corrected.

Page 8, line 278: Please change 'well' to something else, perhaps so the sentence reads '...higher in winter and lower in summer, which is consistent with the seasonal variations of monsoon...'.

Reply:
Corrected.

Page 13, line 378: Change 'due to its being located' to 'due to the site being located'.

Reply:
Corrected.

Page 14, Table 6: Guangzhou is listed twice in this table as being this study but there are two different periods. Perhaps the authors would like to clarify the exact time periods. Is the 2015.1-2015.11 period actually the four months of study during this whole time period and then the 2015.1-2015.2 is one of those four months?

Reply:
We have pointed out whether it is four seasons or one season below the sampling period in Table 6.

Supplement page 7, Fig. S4: The authors may wish to include axis labels for the latitude and longitude.

Reply:
Corrected.

Changes:

Page 7, line 222, i.e., the percent of a species apportioned to the factor.

Page 12, line 354, However, the contribution time-series of LV-OOA (or SV-OOA) can be extracted based on the contribution time-series of the secondary sulfate-rich factor (or the secondary nitrate-rich factor) and the mass percentage of OM in this factor, leaving the remaining mass as the "pure" secondary sulfate (or secondary nitrate). Therefore, a new SOA factor can be reasonably estimated by LV-OOA+SV-OOA.

Page 14, Table 6, (Four seasons) and (Winter).

Page 20, line 583, four months (one for each season).

Supplement, Figure S7, explained variations (% of species).

[revised manuscript text omitted]